# Methods and Guidelines for Metabolism Studies: Applications to Cancer Research

**DOI:** 10.3390/ijms26178466

**Published:** 2025-08-30

**Authors:** Melvin Li, Sarah R. Amend, Kenneth J. Pienta

**Affiliations:** 1Cancer Ecology Center, The James Brady Urological Institute, Johns Hopkins School of Medicine, Baltimore, MD 21287, USA; samend2@jhmi.edu (S.R.A.); kpienta1@jhmi.edu (K.J.P.); 2Pharmacology and Molecular Sciences Program, Johns Hopkins School of Medicine, Baltimore, MD 21287, USA

**Keywords:** cancer metabolism, untargeted metabolomics, Seahorse metabolic flux analysis, isotope tracing, ^13^C-metabolic flux analysis, fluorescent probes, genetically encoded fluorescent biosensors

## Abstract

Metabolism is a tightly controlled, but plastic network of pathways that allow cells to grow and maintain homeostasis. As a normal cell transforms into a malignant cancer cell and proliferates to establish a tumor, it utilizes a variety of metabolic pathways that support growth, proliferation, and survival. Cancer cells alter metabolic pathways in different contexts, leading to complex metabolic heterogeneity within a tumor. There is an unmet need to characterize how cancer cells alter how they use resources from the environment to evolve, spread to other sites of the body, and survive current standard-of-care therapies. We review key techniques and methods that are currently used to study cancer metabolism and provide drawbacks and considerations in using one over another. The goal of this review is to provide a methods’ guide to study different aspects of cell and tissue metabolism, how they can be applied to cancer, and discuss future perspectives on advancements in these areas.

## 1. Introduction

Metabolic reprogramming is one of the critical hallmarks of cancer [1]. Just over 100 years ago, Otto Warburg identified that cancer cells make a metabolic switch to break down glucose through aerobic glycolysis rather than oxidizing it in the tricarboxylic acid (TCA) cycle—coined the “Warburg effect” [2,3,4]. From this initial discovery, Warburg postulated that since glycolysis was the preferred pathway to break down glucose for energy, cancer cells were deficient in mitochondrial respiration and hypothesized that cancer initiation occurred through respiration defects [3,4]. Although this has been disproven, this work pioneered the understanding of how metabolism affects cancer progression. Cancer cells require vast amounts of resources to generate building blocks for growth and proliferation [5]. They alter many metabolic pathways to increase biomass and energy production and adapt to environmental stressors. How cancer cells alter their metabolism changes over disease progression as they are exposed to different environments in the primary tumor, in the bloodstream during metastasis, and in the secondary site after extravasation and colonization [6].

Cancer cells have also been shown to reprogram their metabolism in response to various standard-of-care therapies such as chemotherapy and radiotherapy [7,8,9]. The goal of these anti-neoplastic treatments is to induce DNA damage and/or overwhelm the intrinsic stress response system in cancer cells to cause cell death [10]. Cancer cells that are metabolically flexible can adapt to the stresses induced by those agents and survive, leading to selection for resistant populations [8]. As our understanding of metabolic reprogramming has evolved over the past century through advancements in molecular biology, biomedical engineering, imaging sciences, and analytical chemistry, it has also revealed that tumors in vivo are heterogenous in metabolic phenotypes, making it difficult to develop therapeutic strategies to target metabolic pathways [6]. Understanding how the metabolism of the tumor changes at the cellular level and at the tissue level in response to current standard-of-care treatments can address the issue of metabolic heterogeneity and provide better therapeutic strategies to target cancer metabolism.

Studies use a wide range of methods to probe different areas of cellular and tissue metabolism, making it difficult to know where to start. The goal of this review is to provide a methods’ guide to study different aspects of cell and tissue metabolism, and how they can be applied to cancer, and discuss future perspectives on advancements in these areas. We discuss complementary techniques, including fluorescent dyes, genetically encoded fluorescent biosensors, Seahorse real-time metabolic flux assays, and metabolomics, isotope tracing, and metabolic flux analysis. We have outlined how these methods can best suit one’s research depending on the goals of the project and their potential drawbacks. Throughout this review, we will reference Figure 1 as a roadmap and decision tree for the order in which methods are discussed.

## 2. Untargeted Metabolomics/Lipidomics

Untargeted metabolomics is a powerful method to obtain an overview of the global metabolic profile and provides preliminary data to generate hypotheses and ask new questions (Figure 1). Untargeted metabolomics can provide absolute or relative metabolite levels between groups intracellularly or extracellularly [12,13]. Metabolites are substrates and products of enzymatic reactions in pathway networks, ultimately leading to the production of energy and macromolecules for various cellular processes [13]. Metabolomics is a mass-spectrometry-based method to analyze metabolite abundances from a complex mixture by their mass-to-charge ratios (*m*/*z*) [14]. The typical workflow for metabolomics involves sample harvesting, metabolite extraction, chromatography, ionization, fragmentation, and detection. Samples such as cells, tissue, blood, saliva, or urine can be used for metabolomics analysis. Metabolites are extracted with a cold organic solvent such as acetonitrile that stops all enzymatic activity, allowing for metabolism to be quenched [13]. This ensures that enzymes do not break down any metabolites in the process of sample preparation. After extraction, the mixture is still very complex, and it may be hard to identify metabolites if it is just run on the mass spectrometer right away. Separation via chromatography is an essential step to determine metabolites that have similar *m*/*z* ratios, such as metabolites that are isomers of one another [13,14]. Metabolites are not created equally—some have polar functional groups and some are nonpolar, while some are volatile [15,16]. Chromatography can take advantage of the inherent physical and chemical differences in metabolites to separate different classes to run on the mass spectrometer [13,14]. The two common separation techniques used for mass spectrometry are liquid chromatography (LC) and gas chromatography (GC) [13,14,15,16]. LC uses a liquid in the mobile phase of the separation column, while GC uses a gas [15,16]. In LC, polar metabolites will elute at specific retention times in one column, and nonpolar metabolites will elute at a different retention time in that same column [14]. In GC, samples are heated to a very high temperature and metabolites are separated by their volatility, which is the temperature at which they vaporize into a gas [16]. Once separated, the fractions can be ionized and fragmented in the mass spectrometer for detection of *m*/*z* for each metabolite [13]. The area of the detected *m*/*z* peaks is proportional to the abundance or concentration of a metabolite [13]. To determine the concentration or absolute amount of a metabolite within cells or tissues, a standard curve must be generated using known concentrations of each metabolite.

To interpret data from an untargeted metabolomics experiment, we recommend visualizing the dataset in terms of clusters, pathways, and networks instead of only plotting the concentrations of individual metabolites. The strength of untargeted metabolomics lies in the plethora of downstream analyses and visualizations to assess global changes in the metabolome [13]. One useful tool that does not require any bioinformatics experience to perform these analyses is MetaboAnalyst 6.0 [17]. This web-based analysis platform has an easy-to-use graphical interface to upload and normalize data, perform statistical analyses, and generate graphs such as volcano plots, enrichment plots, and heatmaps. As shown in Figure 2A, a dimensionality reduction analysis was performed using a publicly available dataset by Tang et al. (published under a Creative Commons Attribution 4.0 International License) to visualize the separation of each classified group (estrogen receptor-negative vs. estrogen receptor-positive) based on components determined by the concentrations of all the metabolites [18]. In addition, volcano plots can be generated to obtain an overview of enriched metabolites in one group over another based on log2 fold changes and statistical significance. As shown in Figure 2B, there were about 50 enriched metabolites in the tumor tissue of patients with estrogen receptor-negative (ER−) breast cancer when compared to tumor tissue of patients with estrogen receptor-positive (ER+) breast cancer. To view how the changes in metabolite concentrations can affect how samples cluster with one another, a heatmap with a dendrogram can be used, with a color gradient that corresponds to the magnitude fold change in metabolite concentrations (Figure 2C).

In addition to visualizing overall changes in the metabolome, MetaboAnalyst can extract biologically meaningful information from the data by grouping metabolites into known pathways from databases, such as the Kyoto Encyclopedia of Genes and Genomes (KEGG), and interpret the functional change in a pathway based on the changes in metabolite concentrations between each group [17,19]. Similar to Gene Set Enrichment Analysis (GSEA), MetaboAnalyst can perform Metabolite Set Enrichment Analysis (MSEA) and produce an enrichment plot that lists the top 25 pathways that differ between two groups based on an enrichment ratio and statistical significance [17]. Based on the MSEA shown in Figure 2D, ER-negative breast cancer tissues exhibit an enrichment in pathways involved in pyrimidine and amino acid metabolism when compared to ER-positive cancer tissues. The changes in concentration of the specific metabolites that contribute to the MSEA and pathway analysis can also be viewed (Figure 2E). For scientists that have bioinformatics experience, MetaboAnalyst can be coupled with the MetaboAnalystR package to customize statistical analyses and plotting results [20]. These analyses can help drive hypothesis generation and allow researchers to ask new questions to dive deeper into the top pathways that were significantly enriched. Untargeted metabolomics can be integrated into multi-omics analysis for a system-wide view of changes between groups based on the genome, transcriptome, proteome, and metabolome. This approach allows the researcher to unravel the complex biological processes ranging from genotype to phenotype. Benedetti et al. integrated matched transcriptomic and metabolomics data across 11 tumor types from 15 datasets and measured a gene–metabolite covariation statistic to assess the correlation of gene expression changes and metabolite abundance changes [21]. For a comprehensive list of other bioinformatics data analysis guidelines and tools for untargeted metabolomics, refer to the review by Chen et al. [22].

Through untargeted metabolomics, Amrutkar et al. profiled the metabolome of tumor tissue from pancreatic cancer patients that either never received treatment or received neoadjuvant fluorouracil, leucovorin, irinotecan, and oxaliplatin (FOLFIRINOX) therapy [23]. They observed an enrichment in purine metabolism and serine metabolism pathways in the neoadjuvant treatment group over the treatment-naïve group [23]. Potential biomarkers of treatment response were identified as AMP, CMP, UDP-glucose, UDP-GalNac, as they exhibited a positive correlation with a decrease in the pancreatic cancer biomarker carbohydrate antigen 19-9 (CA 19-9) [23]. Lyu et al. identified distinct metabolic profiles of non-small-cell lung cancer tissue (NSCLC) in patients that also had diabetes mellitus when compared to patients with just non-small-cell lung cancer. Patients with NSCLC and diabetes mellitus had increased levels of putrescine, carnitine-C12, biotin, and cytosine in their tumor tissues. This study revealed the potential metabolic comorbidity of having diabetes mellitus in the progression of NSCLC [24]. Another study by Moreno et al. used metabolic profiling to reveal a therapeutic vulnerability in nucleotide metabolism in human lung tumors [25]. This method has also been used to identify potential biomarkers from the sera and urine of cancer patients [26,27]. Nizioł et al. performed untargeted metabolomics on blood serum samples from 100 patients with bladder cancer and compared it to 100 non-cancer controls [27]. Through principal component analysis (PCA), the bladder cancer group was separated from the non-cancerous group based on the levels of serum metabolites [27]. In addition, they were able to separate a high-grade bladder cancer group from the low-grade bladder cancer group using the same analysis. They reported that the class of metabolites that differentiated the patients with bladder cancer and the non-cancerous group was lipid metabolites [27]. There were 38 metabolites that were observed to differ between non-cancer controls, low-grade bladder cancer, and high-grade bladder cancer [27]. This study presents potential biomarkers for the early detection of bladder cancer, which can aid in better prognosis for patients.

### Drawbacks and Considerations for Untargeted Metabolomics/Lipidomics

One drawback of untargeted metabolomics/lipidomics is the depth and range at which metabolites can be detected. These methods are limited in their ability to detect lower abundance metabolites, which may be of importance in a biological context [28]. In addition, one cannot use reference standards for unknown metabolites in a discovery experiment, limiting the user to only use detect relative abundances of metabolites between two groups [29]. Another drawback is the limitation of inferencing effects on pathways solely based on metabolite concentrations. An increase in metabolite concentration could result from different pathway fluxes that contribute to the production and consumption of that metabolite (refer to the Isotope Tracing and Metabolic Flux Analysis section for more detail). Once a hypothesis has been generated from a discovery untargeted metabolomics experiment, the next step is to decide if the metabolic phenotype of interest can be tested through extracellular or intracellular measurements (Figure 1). If the phenotype involves changes in oxidative phosphorylation or glycolysis, we recommend starting with the Seahorse real-time metabolic flux analysis to measure extracellular fluxes of groups of interest (Figure 1).

## 3. Seahorse Real-Time Metabolic Flux Assays

One of the standard techniques in the field to assess cellular metabolism is Seahorse real-time cell metabolic analysis. This bioanalyzer from Agilent Technologies measures extracellular fluxes to capture the oxygen consumption rate (OCR) as a readout of mitochondrial function, and the extracellular acidification rate (ECAR) as a readout of glycolytic activity of one’s cells of interest in a plate-based format. The assays are available in a 24-well plate and 96-well plate format, allowing users to perform high-throughput screening experiments assessing bioenergetics with multiple conditions and cell lines.

There are two commercially available assays to specifically assess mitochondrial respiration and glycolytic activity in cells: the Mito Stress test and the Glycolysis Stress test, respectively. The Mito Stress test involves measuring OCR while subjecting cells to various mitochondrial-targeting drugs, such as oligomycin, FCCP, and a mix of rotenone and antimycin A, to identify the different capacities of respiration at specific timepoints (Figure 3A). Before the treatments begin, the basal OCR is observed, representing the endogenous respiration rate of the cells. As cells are treated with oligomycin, the OCR drops as the mitochondrial ATP synthase is inhibited [30]. The difference between the basal OCR and the OCR in response to oligomycin treatment represents ATP-linked respiration [31,32]. Cells are subsequently treated with carbonyl cyanide 4-(trifluoromethoxy)phenylhydrazone (FCCP), an uncoupling agent that dissipates the proton gradient generated by the electron transport chain (ETC) [33]. As the proton gradient decreases, the cells will increase activity across the ETC to try to maintain their mitochondrial membrane potential for oxidative phosphorylation, ultimately maximizing their OCR [31,32]. The difference in the basal respiration rate and the maximum respiration rate represents the respiratory reserve capacity of the cells. Lastly, treatment with rotenone (Complex I inhibitor) and Antimycin A (Complex III inhibitor) will shut off the ETC before electrons reach Complex IV (where oxygen is consumed) and quickly drop the OCR to below basal levels and reveal respiration that occurs outside of the mitochondria [31,32]. One can also measure the ECAR during the Mito Stress test to assess how glycolysis and CO_2_ production (refer to drawbacks and considerations in this section) are affected by the addition of the mitochondrial-targeting drugs [31].

The Glycolysis Stress test involves measuring ECAR, as lactate is produced and released, while culturing cells in glucose, oligomycin, and 2-deoxyglucose (2-DG) to identify different glycolytic capacities at specific timepoints (Figure 3B). First, cells are starved of glucose to exhibit low ECAR. The addition of glucose stimulates cells to become glycolytic, while oligomycin inhibits mitochondrial ATP synthase, shifting cells to maximize ATP production through glycolysis. The cellular response to oligomycin measured through the increase in ECAR represents the maximum glycolytic capacity of the cells. The difference in ECAR between the maximum glycolytic rate and the basal glycolytic rate represents the glycolytic reserve of the cells. Treatment with 2-DG afterwards will rapidly decrease the glycolytic rate by competing with glucose to bind hexokinase, which is responsible for the first rate-limiting step in glycolysis by phosphorylating glucose to glucose-6-phosphate [37,38]. As mentioned previously, once 2-DG is phosphorylated, it cannot be broken down further with downstream glycolytic enzymes and ultimately inhibits the glycolytic pathway [37,38]. This last treatment acts as a control for the experiment to ensure that the effects observed are from glycolytic activity. One can also measure the OCR during the Glycolysis Stress test to obtain information on how mitochondrial respiration is affected during the addition of glucose, oligomycin, and 2-DG.

Once data is collected, it is important to normalize the values to accurately compare between groups. Depending on the question at hand, researchers in the field have normalized their OCR and ECAR values to cell number, total protein, mitochondrial content, and DNA content [39,40,41,42,43]. Once normalized, OCR and ECAR values that resulted from each drug treatment can be plotted. For example, the maximal respiration rate, basal respiration rate, and respiratory reserve can be compared between two groups [31]. In addition to plotting OCR and ECAR values, mathematical conversions can be performed to obtain the ATP production rate from glycolysis and mitochondrial respiration in units of picomoles per minute [44]. This can be useful in directly assessing energetic changes in one condition over another and linking ATP demand to metabolic reprogramming [44]. For information on the equations to convert OCR and ECAR measurements to ATP production rate and the methods used to generate those equations, refer to the work of Desousa et al. [44].

Another way to interpret the data is by plotting OCR by ECAR to obtain an energetic profile of the cells of interest. The plot can be split into four quadrants: low OCR–low ECAR (low energetic), high OCR–low ECAR (aerobic), low OCR–high ECAR (glycolytic), and high OCR–high ECAR (energetic). One can then observe how the profile changes in those four quadrants as cells are manipulated with pharmacological agents or with specific gene knockdowns or knockouts (Figure 3C). It can also be used to compare between cancer cell lines to identify distinct metabolic phenotypes that can be leveraged for targeting. For example, Guha et al. plotted OCR vs. ECAR of triple-negative breast cancer (TNBC) cell lines and non-TNBC cell lines and found that TNBC cell lines exhibit decreased mitochondrial respiration and are more glycolytic, suggesting that the more aggressive cell lines have mitochondrial defects and shift towards glycolytic metabolism [41]. In addition, Lanning et al. reported differential energetic phenotypes within the TNBC group of cell lines, where MDA-MB-231 cells exhibited a higher glycolytic phenotype and lower capacity for mitochondrial respiration than MDA-MB-468 cells, while HCC70 cells exhibited elevated mitochondrial respiration and glycolysis when compared to all other TNBC cell lines [40]. When treating those cell lines with various metabolic inhibitors, the authors found that MDA-MB-468 cells were more sensitive to metformin (Complex I inhibitor) than MDA-MB-231 cells, and HME1 and HS578T cells were not affected by metformin as they exhibited a highly glycolytic and low-respiration phenotype [40]. Moreover, the HME1 and HS578T cells were more sensitive to 2-DG and 6-aminonicotinamide (6-AN), which inhibit glycolysis and the pentose phosphate pathway, respectively [37,38,40,45].

One advantage to the Seahorse real-time metabolic analysis is the ability to multiplex with other assays. Little et al. presents a multiplex assay that involves nuclear staining, mitochondrial staining, and mitochondrial function staining downstream of obtaining OCR and ECAR measurements with the Seahorse metabolic flux analyzer [42]. After assessing mitochondrial bioenergetics, cells are stained with a nuclear dye for data normalization as a proxy for cell count and other fluorescent dyes of interest (i.e., MitoSox, DCF-DA, TMRE, MitoTracker Green), followed by high-content imaging and image analysis. This allows the user to obtain multiple metabolic readouts from one experiment that are linked to one another. For example, in addition to data normalization, one can perform cell cycle analysis based on nuclear stain intensity as a readout for DNA content to correlate in which phases of the cell cycle are mitochondria most active, and which phases are more reliant on glycolysis [42]. With high-content imaging and staining with various mitochondrial dyes, one can also assess mitochondrial morphology, content, and membrane potential downstream of OCR measurements. The authors identified that TMRE fluorescence exhibited a strong positive correlation with OCR in T3M4 cells, which is expected since the mitochondrial membrane potential is generated during oxidative phosphorylation [42,46]. In addition, as a proof-of-concept experiment, they treated cells with Antimycin A (Complex III inhibitor) and observed increased mitochondrial fragmentation and decreased OCR; through many other studies, it is known that fragmented mitochondria exhibit lower oxidative capacity [43,47,48,49,50]. For more information on the methodology and how to perform these assays in vitro, these articles provide detailed protocols across various cell types [51,52], while Divakaruni et al. and Yoo et al. provide insights into the analysis and interpretation of the data generated from these assays [31,32,53].

This approach is not limited to cell lines and can also be applied to organoids and tissues ex vivo. Studying the metabolism of cancer-derived organoids and tissues can better recapitulate the metabolic phenotype of a tumor that is observed in vivo. The three dimensional (3D) structure creates more heterogeneity in nutrient availability, waste disposal, pH, and oxygen that cells in a tissue would experience in the body [54]. Tidwell et al. and Russel et al. examined mitochondrial respiration and glycolytic activity of cancer spheroids using this technology and found that they were vastly different from their two-dimensional (2D) adherent cell line counterparts [55,56]. Frederick et al. observed increased mitochondrial function and reduced glycolytic activity in metastasizing spheroids when compared to adherent cells in ovarian cancer [57]. In addition, Compton et al. identified that ovarian cancer spheroids increased lactate excretion and exhibited low respiration during adhesion, suggesting that metabolic reprogramming occurs in ovarian cancer metastasis during colonization of a secondary site [58]. When studying esophageal tumor patient biopsies ex vivo, Buckley et al. identified that treatment-naïve tumor tissue exhibited a high respiration rate and a low glycolytic rate, leading to a therapeutic sensitivity to pyrazinib treatment [59]. In all these studies, the authors stress the importance of using more complex model systems when evaluating tumor metabolism to identify therapeutic targets [55,56,57,58,59]. Published protocols, such as those of Campioni et al. and Ludikhuize et al., provide optimized workflows to perform the Seahorse extracellular flux assays in organoids [54,60], while these articles provide strategies to retain OCR and ECAR in various tissues for those assays [61,62,63,64,65].

Acin-Perez et al. recently developed a method to apply Seahorse metabolic flux analysis to frozen tissues, making it possible to obtain metabolic data from tissue biobanks [66]. Freeze–thawing tissues causes permeabilization of the mitochondrial outer membrane, leading to the leakage of respiration substrates such as succinate, pyruvate, and NADH, and collapsing the respiratory capacity of mitochondria in frozen tissues [66]. In their protocol, tissues are homogenized and centrifuged to obtain the post-nuclear fraction. The authors demonstrated that the post-nuclear fraction of frozen tissue retained respiratory capacities using their protocol, allowing researchers to reduce the amount of tissue to use for experiments to 200–500 milligrams instead of the conventional standard of 500–2000 milligrams [66]. Once the post-nuclear fraction is obtained, protein concentrations were measured, and an equal amount of homogenate protein was loaded into the Seahorse assay plate. The protocol is designed to stimulate and measure the activities of Complexes I, II, and IV through the addition of various substrates. In the first port injection of the Seahorse assay, samples were injected separately with NADH as a substrate for Complex I, or a combination of succinate as a substrate for Complex II and rotenone to inhibit Complex I [66]. After OCR is measured from the first injection, the samples are then injected with a combination of rotenone and Antimycin A (like the conventional Seahorse assay to inhibit Complexes I and III), which should drop the OCR. The third injection contains *N,N,N’,N’*-tetramethyl-*p*-phenylenediamine (TMPD) and ascorbic acid, both of which activate Complex IV to maximize respiration. The last injection contains azide, which inhibits Complex IV activity. For further details on the protocol, refer to the publication as it outlines the exact concentrations and mix and measure times for each substrate [66]. Adapting this technique, Sarver et al. reported a mitochondrial respiration atlas to assess mitochondria function across sex and age comparing 33 frozen tissues from different human organs [67]. Interestingly, the authors found that older males exhibited increased mitochondrial function in skeletal muscles and heart muscle while also having decreased mitochondrial function in brown adipose tissue and the brain when compared to younger males [67]. This freeze–thaw technique unfortunately does not retain the glycolytic capacity of the tissue and cannot report ECAR and thus can only be used to measure OCR. Nevertheless, this is a powerful tool to use to probe mitochondrial function from tissue biobanks with a lower amount of input tissue, allowing researchers to test clinically relevant samples and draw conclusions that can impact the biological understanding of indicated diseases and potential treatments for those diseases.

### Drawbacks and Considerations for Seahorse Real-Time Metabolic Flux Assays

One technical note to consider is that glycolysis is not the only contributor to the acidification of the media. Throughout the TCA cycle, CO_2_ is produced through decarboxylation of isocitrate and alpha-ketoglutarate [68]. Once CO_2_ is released from the cells and into the cell culture medium, it is hydrated and subsequently dissociated into bicarbonate and a proton [31,69]. The proton contributes to the acidification of the media, affecting the ECAR readout. Luckily, the drug treatments performed in the Glycolytic Stress test will help determine any non-glycolytic acidification that contributes to the ECAR (Figure 3B). In addition, it is imperative to perform a titration of all the compounds needed to assess different aspects of OCR and ECAR before conducting an experiment comparing two groups. Different cell lines have different tolerances to ETC-disrupting drugs depending on the state of their mitochondria [70,71]. Testing different cell densities should also be performed to optimize signal detection and assess the effects of confluency on the metabolism of one’s cells [71,72]. Although the Seahorse metabolic flux analyzer is a powerful tool to assess metabolic profiles in cells and tissues, it is limited by solely measuring extracellular fluxes (Figure 1). If one is interested in mapping specific intracellular metabolites and fluxes, we recommend performing isotope tracing and metabolic flux analysis (Figure 1).

## 4. Isotope Tracing and Metabolic Flux Analysis

While untargeted metabolomics provides a top–down view of the metabolome, pathway activity cannot be interpreted solely based on metabolite concentrations. An increase in metabolite concentration could result from an increased rate of production or a decreased rate of consumption of that metabolite [13]. Lowry et al., Xu et al., and Jeremy et al. reported discrepancies in metabolite concentrations and flux through a pathway [73,74,75]. In addition, there are many pathways that affect the concentration of a specific metabolite, and isotope tracing is a technique that can be utilized to dissect which pathway contributes to the increase or decrease in the metabolites of interest [12,13,76]. For example, citrate can be produced from a variety of different carbon sources, such as glucose, glutamine, and aspartate, each metabolized through different pathways. After glycolysis, pyruvate can be decarboxylated to acetyl-CoA and react with oxaloacetate to make citrate [77,78]. In contrast, anaplerosis of glucose-derived pyruvate can replenish the oxaloacetate pool to also make citrate [12,79,80,81,82]. Citrate can also be produced through glutamine oxidation in the TCA cycle or through reductive carboxylation of glutamine [83,84,85]. If the goal is to identify and inhibit specific pathways or enzymatic activities that cancer cells enhance to survive and proliferate, it is imperative to assess the fluxes of those pathways to define a target.

Isotope tracing involves introducing a nutrient source with atoms containing one or more heavy, non-radioactive isotopes for cells to metabolize, such as [U-^13^C] glucose, where all six carbons are ^13^C. As the heavy-nutrient source, or “tracer”, is utilized and broken down through various pathways, the heavy atom is incorporated into the metabolites of those pathways [12]. One can assess the fractional contribution of the tracer to the metabolites of targeted pathways to infer how one group uses a nutrient when compared to another group [12]. There are various tracers commonly used to assess the activity of different pathways. For example, [U-^13^C] glucose is mainly used to study glycolysis and glucose contribution to the TCA cycle, while positionally labeled glucose such as [1,2-^13^C] glucose is used to compare the relative activity between the pentose phosphate pathway and glycolysis [86,87]. In addition, [U-^13^C] glutamine is commonly used to probe glutamine oxidation through the TCA cycle, as well as reductive carboxylation for fatty acid synthesis [84,88]. ^15^N tracers are available to track amino acid synthesis and nitrogen metabolism (i.e., nucleotide biosynthesis, urea cycle, polyamine pathway, and hexosamine pathway), while ^2^H tracers can be utilized to study pathways involving hydrogen transfers such as NADPH and fatty acid metabolism [89,90,91,92,93]. For an extensive table indicating which specific tracers can be used to probe different pathways, refer to the work of Jang et al. [13].

After selecting a tracer to probe a pathway of interest, the labeled substrate is added to media at an optimized concentration, usually at the same concentration of the unlabeled nutrient. Typically, the media are free of the unlabeled substrate; so, once the tracer is added, 100% of that substrate is from the tracer itself. Pilot experiments should be performed to optimize the timing of tracer-containing media incubation on cells. The purpose of optimization is to determine the timepoint at which isotopic steady state is reached, i.e., when the labeling of metabolites in a pathway is constant over time (Figure 4A). The labeling kinetics are determined by the flux through each reaction of the pathway and the pool size of each metabolite [88]. Thus, if the labeling dynamics are constant over time, the flux of the reaction can be calculated, assuming the pool size of each metabolite does not change. For these pilot experiments, we recommend assessing the labeling of the metabolites in the pathway of interest at multiple timepoints (0, 6, 12, 24 h) as different pathways will incorporate the tracer at different rates to reach an isotopic steady state. For example, an isotopic steady state is reached with glucose tracers in glycolytic metabolites within minutes, while TCA cycle metabolites need several hours [12,88].

The incorporation of heavy atoms into downstream metabolites can be detected with three main analytical methods: LC-MS, GC-MS, and nuclear magnetic resonance (NMR) spectroscopy. Since mass spectrometry measures ratios of mass to charge (*m*/*z*), it can detect mass shifts in a metabolite caused by incorporation of the heavy atom with high resolution [96]. For a metabolite with a mass m, the unlabeled metabolite will have a mass shift of m+0, while the incorporation of one heavy atom would produce a peak at m + 1, and so on. Identifying the different masses of each metabolite that occur from incorporation of the heavy atoms (m + 0, m + 1, m + 2…) will produce a mass isotopomer distribution (MID) [12]. Comparing the MIDs of detected metabolites between two groups is useful to determine which pathways contribute to breaking down the substrate in one group over another. In addition, chromatography performed on the sample prior to mass spectrometry analysis allows for better separation and identification of metabolites from a complex mixture. NMR, on the other hand, does not measure *m*/*z* ratios, but detects changes in the nuclear spins of the atoms as they are exposed to a magnetic field [97]. The nuclear spins are affected by neighboring atoms, leading to spectra patterns generated by scalar coupling and allowing for the detection of different chemical structures based on those spins [98]. The spins of nuclei in heavy isotope-labeled metabolites are distinct from the nuclei spins of unlabeled metabolites. One can measure the percent incorporation of the tracer by comparing the NMR spectra of the labeled metabolite and the unlabeled metabolite [96]. One advantage of NMR-based isotope tracing analysis over mass spectrometry is the ability to detect the position of the heavy atom in the metabolite’s chemical structure [96,98]. This information is useful in tracing where the heavy atom goes in each metabolite and identifying mechanisms of specific reactions. With positional labeling information, one can also calculate how many heavy atoms were incorporated and generate a MID for each metabolite (m + 0, m + 1, m + 2…). For an in-depth workflow on performing isotope-tracing studies using NMR, refer to the work of Saborano et al. and Lin et al. [76,99].

One important data-processing step after obtaining data of the MIDs of each metabolite is to correct for the natural abundance of stable isotopes. ^13^C, ^2^H ^15^N, ^17^O, ^18^O, and ^34^S naturally exist in the environment and contribute to a metabolite’s mass shift when detected via mass spectrometry [12,100]. If the natural abundance of stable isotopes is not corrected for, it may skew the interpretation of the isotope-tracing data since the mass isotopomer distributions will contain mass shifts from one’s stable isotope tracer, as well as the natural abundance of stable isotopes of that atom [101]. For example, the fate of the [1,2-^13^C] glucose tracer results in m + 1-labeled or m + 2-labeled glycolytic metabolites depending on its trajectory into the pentose phosphate pathway (PPP) or into glycolysis [12,13,88,102]. Regardless of the incorporation of the tracer, the downstream glycolytic metabolites will have an m + 1 peak due to the natural abundance of ^13^C. Thus, to accurately interpret whether the m + 1 peak resulted from the environment or flux through the PPP, one must correct for the natural abundance of ^13^C [13]. In addition, the metabolites of interest contain atoms that have natural abundances of their respective stable isotopes, and it is recommended that those be corrected for as well since they will also contribute to the mass shift [103,104]. There are many published methods on natural abundance corrections available, utilizing different mathematical approaches and different programming languages (Table 1). Some methods are compatible with experiments that use multiple tracers, such as ^13^C-^15^N and ^2^H-^13^C tracers, while some methods can only correct data from experiments using one tracer, such as ^13^C and ^2^H.

One limitation of assessing metabolic pathway activity solely through mass isotopomer distributions (MIDs) is the ability to resolve the fluxes of cyclic, reversible and exchange reactions [102]. The complexity with which atoms are re-arranged in some metabolic pathways, such as the pentose phosphate pathway (PPP) and the TCA cycle, makes data interpretation difficult for those specific pathways [88]. In addition, MIDs will provide qualitative data as these only report on the relative contribution of a pathway to producing a metabolite and do not represent absolute fluxes of the pathway [12]. One can perform more robust analysis through mathematical modeling. A computational approach for analyzing metabolic fluxes can simulate reactions and estimate flux values from the experimental data quantitatively. ^13^C-metabolic flux analysis (^13^C-MFA) integrates the extracellular fluxes, such as glucose and glutamine consumption rates and lactate excretion rate, with the MID data from isotope tracing, into a metabolic model to calculate absolute flux values of each reaction in the model [88,102,111]. The metabolic model can be generated using known reaction networks and atom transitions from databases such as KEGG, MetaCyc, and BiGG [112,113,114]. Extracellular fluxes can be calculated by measuring metabolite concentrations in the media over time through LC-MS, GC-MS, plate-based assays, or biochemistry analyzers [102]. The extracellular fluxes constrain the mathematical calculations for the intracellular fluxes and provide absolute units for the model, usually in nmol/h/106 cells [88,102]. With this approach, Britt et al. identified that neutrophils switch to a cyclic PPP to maximize NADPH production when undergoing oxidative burst [87]. With traditional MID analysis, the labeling patterns of the metabolites in the PPP from glucose tracers are too complex to interpret and predict relative fluxes of each reaction due to the reversible nature of the transketolase (TKT) and transaldolase (TALDO1) reactions [102,115]. ^13^C-MFA was able to simulate intracellular fluxes that resembled a cyclic PPP pathway with high reversibility of F6P conversion to G6P, which was validated with knockouts of TKT and TALDO1 [87]. There are many MFA packages available for scientists that do not have any programming experience, such as METRAN (Version 1.0), INCA (Version 2.4), 13CFLUX2 (Version 2.0), and OpenFLUX2 (Version 2.0) [111,116,117,118]. For those who have programming experience and would like to have customizable pipelines for MFA, Python-based packages such as mfapy (Version 0.6.2) and FreeFlux (Version 0.3.6) can be used [119,120].

^13^C-MFA can also address issues in achieving an isotopic steady state in the biological system. There are certain metabolites in pathways where an isotopic steady state cannot be reached due to high exchange fluxes with unlabeled metabolites [12]. For instance, if a system exhibits increased lactate uptake and excretion through MCT1 and MCT4 fluxes, respectively, it may be difficult to achieve steady-state labeling of metabolites downstream of lactate (such as the TCA cycle) from a glucose tracer since high exchange fluxes with unlabeled lactate from the extracellular environment can dilute the tracer-labeling dynamics [12,88,121,122,123]. If one’s biological question or phenotype testing requires a time frame where one cannot achieve isotopic steady state, one can introduce the tracer and take readings at multiple timepoints to assess the labeling kinetics of downstream metabolites and apply isotopic non-stationary metabolic flux analysis (INST-MFA) to estimate intracellular fluxes (Figure 4B) [118,122,124]. The drawback of performing INST-MFA is the need for more data to accurately estimate fluxes since there are more unknowns. Since the experiment is performed under non-steady-state conditions, one must also obtain the intracellular metabolite concentrations and the transient isotope labeling fractions of those metabolites from the tracer [118,124]. INCA (Version 2.4), OpenMebius (Version 1.0), and FreeFlux (Version 0.3.6) are three packages that can perform INST-MFA in addition to steady-state MFA [118,120,124,125].

Through isotope-tracing methodologies in vitro, Rattigan et al. identified that chronic myeloid leukemia cells increase anaplerosis through pyruvate carboxylase (PC) and feed carbons into the TCA cycle, which led to a targeting strategy to inhibit PC and selectively eliminate imatinib-resistant CML cells [80]. In another study, Metallo et al. observed HIF2α-driven reductive carboxylation of glutamine to generate citrate for de novo lipogenesis in A549 lung cancer cells [83]. Isotope tracing has also been used to study cancer metabolism in vivo. Davidson et al. identified, through in vivo tracing studies in mice using [U-^13^C] glucose and [U-^13^C] glutamine, that in *Kras*-driven non-small cell lung cancer models, metabolism of pyruvate through the TCA cycle and anaplerosis is essential for tumor formation in mice, while glutamine metabolism into the TCA cycle is dispensable for tumor growth [126]. Interestingly, these results were not observed in their in vitro cell line models. Proliferation of their cell lines was dependent on glutamine concentration in media, while flank tumors generated with those same cells did not rely on glutamine. Treatment with a glutaminase inhibitor, CB-839, significantly decreased cell proliferation in vitro while having no effect on tumor growth in vivo [126]. In addition, when pyruvate dehydrogenase (PDH) and pyruvate carboxylase (PC) were knocked out, there were no differences in cell proliferation in vitro [126]. When those cells were implanted into mice, however, the tumors did not grow throughout a four-week period. In contrast, when tracing [U-^13^C] glucose in vivo, Kaushik et al. found that patient-derived Von Hippel–Lindau (VHL)-mutant clear cell renal cell carcinoma (ccRCC) xenograft tumors exhibit decreased pyruvate oxidation into the TCA cycle relative to normal kidney tissue [127]. These results recapitulate the metabolic phenotype observed in cell lines and human ccRCC patients. They further characterized the compensatory carbon source used to fuel the TCA cycle and discovered glutamine as the major contributor to that pathway [127]. Treatment with a glutaminase inhibitor, CB-839, or an amidotransferase inhibitor, JHU-083, reduced tumor growth in ccRCC xenograft models [127]. The results from this study showed that patient-derived xenograft ccRCC models represent the metabolic phenotypes observed in human patients while also sharing those phenotypes with respective cell lines. Both studies mentioned here stressed the importance of model selection in evaluating metabolic phenotypes in cancer for targeting [126,127]. While working with cancer cell lines is important, it is imperative to note that they are grown in media with nutrients at non-physiological conditions and they do not experience the same perfusion of nutrients and waste as in vivo tumors do [81,126,127,128,129]. Thus, it may be beneficial to accompany in vitro metabolism data with in vivo tracing experiments when investigating therapeutics against cancer metabolism in the preclinical setting.

There are several completed and currently ongoing clinical trials using isotope tracers, such as [U-^13^C] glucose, [1,2-^13^C] glucose, and [U-^13^C] glutamine, to assess metabolic reprogramming across various tumor types, such as multiple myeloma, leukemia, breast cancer, lung cancer, kidney cancer, brain cancer, and pancreatic cancer [130,131,132,133,134,135,136,137]. In short, patients were continuously infused with the isotope tracer pre-biopsy and throughout the surgery, followed by sample processing of the tumor tissue and blood, and analysis of ^13^C enrichment from extracted metabolites (Figure 4C). In one study performed on patients with triple-negative breast cancer (TNBC), Ghergurovich et al. identified that glucose was the predominant precursor of tumoral lactate, ribose phosphate, and amino acids [132,138]. The relative activity of glucose catabolism pathways by TNBC was previously undefined. By infusing patients with [1,2-^13^C] glucose, they found that most of the ribose phosphate produced is from the oxidative pentose phosphate pathway rather than the non-oxidative pentose phosphate pathway by observing the relative abundance of m + 1 and m + 2 ribose phosphate [138]. In addition, they identified that TNBC tumors also used glucose to fuel the TCA cycle and biosynthesis of amino acids rather than importing their precursors from the serum [138]. They also found a high percentage of ^13^C-labeled lactate in the tumor, suggesting that lactate is primarily produced through glycolysis and is not exchanged with the extracellular space; this metabolic phenotype differs from what is observed in patients with non-small-cell lung cancer [138,139]. In another study, [U-^13^C] glucose was infused in patients with clear cell renal cell carcinoma (ccRCC), which revealed increased glycolytic metabolism when compared to the adjacent healthy kidney tissue while suppressing glucose oxidation in the TCA cycle through decreased pyruvate dehydrogenase activity. These findings were also compared to tracing data from human brain and lung tumors in patients from their previous studies, and interestingly showed that, in contrast with ccRCC tumors, brain and lung tumors preferred to oxidize glucose through the TCA cycle [140]. The results from these studies demonstrate the complex metabolic phenotypes that each tumor type can exhibit in vivo, which can inform physicians and drug development researchers on how to treat patients with those diseases more effectively by targeting the tumors’ reliance on specific pathways. For an in-depth guide on how to perform tracer studies in humans, Kim et al. provides detailed explanations on the methodology and conceptualization of the types of studies that can be carried out, while Faubert et al. provides a step-by-step protocol on performing these experiments [129,141].

### Drawbacks and Considerations for Isotope Tracing and Metabolic Flux Analysis

Unlike untargeted metabolomics, isotope tracing and metabolic flux approaches are more suited to hypothesis testing. Since different tracers provide distinct information on the metabolism of only a certain number of pathways, the selection of the tracer should be performed based on a pathway of interest. Certain studies have performed global isotope tracing, but there are complications with metabolite coverage and capturing tracer enrichment at metabolic and isotopic steady state for every pathway [142,143]. As mentioned previously, a potential drawback of isotope tracing and metabolic flux analysis is the difficulty in achieving steady-state labeling. Conventional ^13^C-MFA software packages, such as METRAN (Version 1.0), assume that isotopic labeling is at steady state. If the incorporation of ^13^C into the metabolites of interest changes over the time period in which one is measuring, it has to be accounted for in the mathematical model [88]. Another consideration is to assess the purity of the tracer that is used in the study. Some tracers are not 100% pure depending on the manufacturer and may contain unlabeled isotopes that may misrepresent the data during MID analysis. Luckily, this tracer impurity can be accounted for when performing natural abundance corrections [103]. Another drawback is the difficulty in resolving compartment-specific fluxes. During sample processing, cells are lysed to obtain the intracellular metabolites, losing compartment specificity since all the metabolites end up in a single pool [88,144]. For example, mitochondrial citrate and cytosolic citrate cannot be discerned since they are mixed during metabolite extraction. This can be accounted for in the metabolic model, indicating that the measured metabolites are from mixed compartments [88,144,145,146]. In addition, one needs prior biological knowledge of the system to decide which tracer to use for the experiment—this can be simulated a priori to assess the optimal tracer for the question at hand [88,111,118]. Metabolic flux analysis is also mathematically complex and may require expertise in bioinformatics, extra experimentation with different tracers, and re-adjustments of the metabolic model to achieve an optimal fit. There have been recent advancements in metabolic flux analysis packages using Bayesian statistics to estimate intracellular fluxes instead of conventional frequentist methods [147,148]. While the main assumption with frequentist statistics for ^13^C-MFA is that there is one “true” flux solution for each reaction that corresponds to the observed data, the Bayesian approach to ^13^C-MFA produces a posterior probability distribution where the flux values may lie for reactions in the metabolic model [149,150]. This allows users to more accurately calculate uncertainties for each flux distribution. For more information on the Bayesian approach to ^13^C-MFA, refer to the work of Theorell et al. to dissect the statistics behind the method, and the work of Backman et al. and Hogg et al. for its application [147,148,149,150].

## 5. Fluorescent Dyes

Fluorescent dyes are chemical tools that are designed to be metabolized by a cell and emit light when excited at a specific wavelength, serving as a proxy for that specific metabolic activity [151]. The chemical nature of the probes allows them to localize to certain compartments of cells and interact with the intended proteins of interest [151]. These dyes can be detected by any instrument that contains the necessary lasers and detectors, mostly by fluorescence imaging, plate readers, and flow cytometry [152]. Both fluorescence imaging and flow cytometry provide readouts with single-cell resolution, allowing the user to capture heterogeneity within a bulk population [151,153]. This is important when assessing tumor metabolism, as not all cancer cells come from the same clone, and cells in different microenvironments of a tissue will exhibit different metabolic phenotypes in vivo [154,155]. One advantage of using fluorescent dyes with imaging systems over flow cytometry is the ability to obtain information on the spatiotemporal dynamics of the metabolic readout, which is useful when studying metabolic compartmentalization at the cellular level, as well as the different microenvironments that can affect metabolism at the tissue level [156,157] (Figure 1). Some of these probes are limited to live cells, while others are well retained and remain fluorescent after fixation, making multiplexing with immunofluorescence possible to detect other markers. In this section, we present various dyes that are used to study glucose import, mitochondrial function, lipid metabolism, and iron metabolism. We also list commercially available dyes that are discussed, and their mechanisms of action and multiplexing capabilities with fixation in Table 2.

### 5.1. Glucose Import

As described by the Warburg Effect, cancer cells take in glucose at a rapid rate when compared to non-cancerous tissue, so much so that this is used as a method to detect tumors in patients through positron emission tomography (PET) with the radioactive probe ^18^F-Fluorodeoxyglucose (^18^F-FDG) [2,3,4,158]. Thus, there have been numerous studies exploring how and why glucose metabolism is dysregulated throughout cancer initiation, progression, and resistance to standard-of-care therapies [159,160,161]. To do this, chemical probes derived from glucose analogs have been developed to evaluate glucose import. 2-(N-(7-nitrobenz-2-oxa-1,3-diazol-4-yl)amino)-2-deoxyglucose (2NBDG) is the most common fluorescent glucose probe used in assessing glucose uptake [162,163]. It is derived from the glucose analog, 2-deoxyglucose (2-DG), where instead of a hydroxyl group at the C2 position, there is a hydrogen one [37]. 2-DG was originally used as a glycolysis inhibitor, where it is phosphorylated by hexokinase once it enters the cell through GLUT transporters but cannot be metabolized any further by any downstream glycolytic enzymes [37,38]. Over time, 2-DG builds up and competes with glucose for hexokinase binding and ultimately inhibits glycolysis [38]. 2NBDG contains a fluorescent moiety at the C2 position instead of a hydrogen one, but surprisingly can be broken down into non-fluorescent downstream metabolites in the glycolytic pathway [162]. Thus, the fluorescence signal detected from 2NBDG represents the net flux of glucose import and glucose breakdown [164]. 2NBDG and its analogs containing different fluorescent moieties have been used extensively for in vitro and in vivo metabolism studies in cancer models. Millon et al. assessed glucose uptake in multiple breast cancer cell lines and used it to monitor their responses to endocrine therapies [165]. Chang et al. observed that high glucose consumption in tumors restricts T cell function in a mouse sarcoma model [166]. In addition, they also showed that standard-of-care immune checkpoint blockades aid in increasing glucose in the tumor microenvironment (TME) to allow T cells to perform glycolysis and improve immunotherapy response [166]. Cai et al. was able to identify circulating tumor cells (CTCs) in multiple mouse xenograft models of breast cancer using 2NBDG fluorescence imaging, resulting in their suggestions for its clinical use to trace CTCs as they are a predictive biomarker of metastatic disease [167].

Although 2NBDG is widely used, it has been previously reported that its uptake kinetics are slower than the uptake kinetics of glucose, potentially due to the increase in molecular size from the fluorescent moiety at the C2 position [164,168]. To circumvent the molecular size issue, Tsuchiya et al. recently developed a click-chemistry labeling method to analyze glucose uptake in vitro, ex vivo and in vivo at a single-cell level [168]. This method involves tagging an azide to galactose (6AzGal) at the C6 position and allowing it to enter the cell via GLUT transporters. A fluorescent reagent containing an alkyne group (BDP-DBCO) is then added onto cells and the azide alkyne cycloaddition reaction occurs, leading to the covalent linkage of 6AzGal and fluorescent reagent inside the cell [168]. This allows for the sugar to be imported into the cell with similar kinetics to glucose while retaining fluorescent detection of the sugar. This method has been validated with glucose competition assays and tested with fluorescent imaging and flow cytometry to capture the metabolic heterogeneity of the groups of interest [168]. One drawback of this technique is the use of galactose instead of glucose. Unlike glucose, galactose is not phosphorylated by hexokinase at the C6 position; this allows for galactose to exit the cell over time since hexokinase phosphorylation of glucose is essential for it to be retained intracellularly [168,169,170]. Thus, this method should be used for rapid measurements of glucose uptake rather than measurements over a long period of time.

One potential drawback of using fluorescence-based glucose uptake dyes is that these do not provide a quantitative readout but a relative readout between two groups as there are no absolute flux units measured. To make this assay more quantitative, radioactive 2-deoxyglucose (2-[^3^H]-glucose) can be used to measure glucose import and obtain an uptake rate in units of nmol/mg/min [171,172]. However, this would require the use of a scintillation counter and working with radiotracers [173].

### 5.2. Mitochondrial Function

Mitochondria are essential organelles that play diverse roles in the bioenergetics, organelle crosstalk, and maintenance of redox status in mammalian cells [174]. Although the Warburg effect suggested that cancer cells had defective mitochondria, this has been disproven [175]. Over the past few decades, many studies have shown that mitochondria play crucial roles in cancer initiation, progression, and resistance to therapies [176,177]. To study mitochondria further, fluorescent probes specific to mitochondria were developed to report on the metabolic function of those organelles. During oxidative phosphorylation, protons are pumped out of the inner mitochondrial membrane and into the intermembrane space as electrons are shuttled through the complexes of the electron transport chain (ETC) [178]. This process ultimately generates a negative membrane potential across the inner mitochondrial membrane [178]. Thus, if mitochondria are more active, this generates a greater negative membrane potential. Fluorescent probes that measure mitochondrial function are designed to be lipophilic and cationic to take advantage of this process and accumulate into active, functioning mitochondria [179]. The fluorescence intensity measured by imaging or flow cytometry is a readout of mitochondrial activity, where higher fluorescence suggests increased mitochondrial function.

The most common fluorescent mitochondrial probes are tetramethyl rhodamine methyl ester (TMRM) (Invitrogen T668; Eugene, OR, USA), tetramethyl rhodamine ethyl ester (TMRE) (Invitrogen T669; Eugene, OR, USA), JC-1 (Invitrogen T3168; Eugene, OR, USA), and MitoTracker dyes (Invitrogen; Eugene, OR, USA) [180]. Although these dyes all accumulate into mitochondria based on the membrane potential, they exhibit distinct properties, and based on the scientific question and experimental goals, one dye would be picked over another. For example, the MitoTracker dyes have a thiol-reactive functional group that reacts with proteins once they enter the mitochondrial matrix, tethering the dye to the proteins [180]. This allows the fluorescent probe to be well retained in the mitochondria regardless of the changes in membrane potential downstream of dye accumulation [180]. If the experimental goal is to capture mitochondrial function at one snapshot in time, then the MitoTracker probes are the recommended dye to use. However, if the goal is to track changes in mitochondrial function over time in presence or absence of drug treatments, then we recommend using TMRM, TMRE, or JC-1. One should consider that each cell may contain differing number of mitochondria; so, the readouts of mitochondrial membrane potential should be normalized to mitochondrial content, cell size, or other markers that scale with mitochondrial biogenesis in the model of interest. MitoTracker Green (MTG) (Invitrogen M7514; Eugene, OR, USA) is reported to be a membrane-potential-independent fluorescent probe that can enter all of the mitochondria in certain cell types [181]. 10-N-nonyl acridine orange (NAO) (Invitrogen A1372; Eugene, OR, USA) has also been shown to be independent of mitochondrial membrane potential in certain contexts [182,183,184], although this is controversial as others have reported its accumulation into mitochondria is affected by membrane potential in their system [185,186]. To ensure it is true in the model system of interest, performing pilot experiments observing fluorescence intensities of MTG or NAO in response to mitochondrial-depolarizing agents would determine whether those dyes are sensitive to mitochondrial membrane potential in that particular system [187]. One can also fix the cells and perform immunofluorescence with a mitochondrial marker, such as TOM20, to normalize to total mitochondrial content [188]. One caveat of this strategy is that TOM20 levels can increase upon stress [189]. In addition, some of these dyes are not compatible for fixation, as shown in Table 2 (found at the end of the document). JC-1 is unique as it can measure the relative polarization state of mitochondria in a cell without the need for normalization to mitochondrial mass [190]. The dye exists in a monomeric state, which emits a green fluorescent signal [190]. As it accumulates into mitochondria via membrane potential, the monomers aggregate together and produce a red fluorescence signal [190]. This process is reversible, where the aggregates can revert back to monomers under mitochondrial depolarization [190]. Thus, the ratio of the red to green fluorescence intensity is a readout of the functional state of mitochondria in a cell that is independent of mitochondrial content or size [190]. Zubaidi et al. utilized JC-1 or TMRM with MTG to map changes in mitochondrial membrane potential in a spatiotemporal context for oocyte maturation and identified that mitochondrial function increases over the 13 h of oocyte development [191].

This experimental design can be applied in cancer metabolism studies to observe how mitochondrial membrane potential changes throughout anti-cancer therapy treatments to identify any resistance mechanisms involved with mitochondrial function. Kuwahara et al. observed that cancer cells resistant to radiotherapy exhibited decreased mitochondrial membrane potential via JC-1 staining [192]. They found that prohibitin-1 (PHB1) was the driver of decreasing mitochondrial function to produce lower levels of reactive oxygen species, and knocking down PHB1 increased mitochondrial membrane potential and reduced radio resistance in their cancer model [192]. These dyes have been integrated into high-content imaging platforms, along with downstream analyses, to allow for high-throughput drug screening to look at different mitochondrial phenotypes in cancer cells in 2D and 3D cultures [193,194].

### 5.3. Lipid Metabolism

In addition to glucose and mitochondrial metabolism, lipid metabolism is also altered in tumors as they progress to metastatic and therapy-resistant disease [195,196,197]. Phospholipids and cholesterol are key components of plasma membranes and double-membrane organelles, which cancer cells need for proliferation [198,199,200]. Lipids such as triglycerides are also a key source of ATP when broken down through fatty acid oxidation (FAO) in the mitochondria [195]. These macromolecules can also be used to generate lipid droplets, which are key organelles in storing neutral lipids to maintain energy homeostasis and prevent peroxidation of certain lipid moieties under oxidative stress [201,202,203]. The most common fluorescent method to track where fatty acids are in the cell is through a BODIPY fluorophore that is covalently linked to a 12-carbon fatty acid (BODIPY 558/568 C12) (Invitrogen D3835; Eugene, OR, USA) [204]. The distribution and fluorescence intensity of the BODIPY fatty acid can be visualized through confocal microscopy once the cells readily take it in at a similar rate as native fatty acid intake [205,206]. Kolahi et al. used BODIPY 558/568 C12 to reveal how lipids are trafficked in cytotrophoblast cells, giving insight into the differential role of lipid droplets in different layers of the human placenta [205]. Rambold et al. developed a pulse chase assay to identify fatty acid localization and lipid droplet lipolysis in cells under starvation, where they initially pulsed BODIPY 558/568 C12 in cells for 16 h and then chased with complete medium or HBSS [207]. They observed that cells chased in a complete medium stored their fatty acids in lipid droplets [207]. Cells chased in HBSS trafficked their lipids into the mitochondria through colocalization of BODIPY 558/568 C12 staining with MitoTracker staining, suggesting that those lipids are oxidized through FAO to make energy under starvation [207].

There are also dyes that accumulate into lipid droplets, such as NileRed (Invitrogen N1142; Eugene, OR, USA), Lipi-Blue (Dojindo Laboratory LD01; Kumamoto, Kyushu, Japan), Lipi-Green (Dojindo Laboratory LD02; Kumamoto, Kyushu, Japan), Lipi-Red (Dojindo Laboratory LD03; Kumamoto, Kyushu, Japan), and BODIPY 493/503 (Invitrogen D3922; Eugene, OR, USA) [208,209,210]. Nile Red is a solvatochromic dye that only becomes fluorescent when exposed to a hydrophobic environment [211]. When binding to a polar lipid species, Nile Red will emit a red fluorescence signal; however, when binding to a neutral lipid species, Nile Red will emit a yellow-orange fluorescence signal [212]. This allows the user to differentiate between lipid moieties in the cell [210]. Lipi-Blue, Lipi-Green, and Lipi-Red were developed to specifically label lipid droplets and no other lipid species [210]. Kostecka et al. identified that cancer cells surviving chemotherapy exhibited increased levels of lipid droplets using NileRed staining and was able to increase sensitivity to cisplatin when knocking down or pharmacologically inhibiting DGAT1, a key protein responsible for lipid droplet formation [213]. Through Nile Red staining, Pilliai et al. observed that breast cancer cells generate lipid droplets from ketogenic amino acids to survive in acidic pH, which is a common feature of the tumor microenvironment in vivo [214].

To detect the breakdown of fatty acids, Uchinomiya et al. developed Probe 10, which only becomes fluorescent when metabolized through FAO [215]. The probe becomes more fluorescent as it is sequentially oxidized in the pathway, allowing the user to detect FAO flux over time [215]. There are also fluorescent dyes that were generated to detect oxidized lipid moieties. Ferroptosis, a non-apoptotic form of cell death that is characterized by the peroxidation of lipid membranes, has been gaining traction in the field as a way to induce cell death in cancer [216,217,218]. The most common fluorescent probes to detect lipid peroxides are diphenylhexatriene (DPH), Liperfluo (Dojindo Laboratory L248; Kumamoto, Kyushu, Japan), and BODIPY-C11 (Invitrogen D3861; Eugene, OR, USA) [219]. These dyes localize to lipid membranes and become oxidized when interacting with the free radical species on the lipid [219]. When oxidized, the dye emits a fluorescent signal that is a relative readout of lipid peroxidation in a cell [219]. BODIPY-C11, similar to JC-1, is a ratiometric dye, where the fluorescent color changes from red to green under oxidation, allowing the user to quantify the proportion of oxidized lipid membranes to undamaged lipid membranes [220,221]. Through BODIPY-C11 staining, Loftus et al. observed that cancer cells surviving chemotherapy converged to a resistant cell state that contains increased levels of lipid peroxides, sensitizing them to GPX4 inhibition and ultimately, ferroptosis [222]. In addition, Zeng et al. identified that neutrophils stained positive for oxidized BODIPY-C11 undergo ferroptosis, leading to increased immune suppression and chemoresistance in breast cancer [223].

### 5.4. Iron Metabolism

Iron is a crucial element that plays diverse roles in many biological processes to facilitate electron transport [224,225]. It acts as a co-factor for many enzymatic reactions and DNA polymerases while also being a key component in iron–sulfur (Fe-S) clusters that are present in the inner membrane of mitochondria [225,226,227,228]. Since cancer cells have increased energetic and catalytic needs for DNA replication, proliferation and metastasis, they require a high amount of iron, leading to alterations in the uptake, storage, and turnover of iron and its regulatory proteins [225]. Iron chelators have been repurposed to bind to iron in cancer cells with high affinity and render them non-reactive [229]. However, many of these compounds either had short half-lives, exhibited poor bioavailability, or did not produce partial or complete response in patients in clinical trials [229]. While an increase in iron can be beneficial for cancer cell proliferation, survival, and metastasis, it can also reveal potential therapeutic vulnerabilities if the iron pool is dysregulated. Labile iron, Fe^2+^, is highly reactive and is one of main drivers of lipid peroxidation in ferroptosis through the Fenton reaction [216,218,230]. Labile iron can be trafficked and stored in ferritin as ferric iron (Fe^3+^), which is more stable and less reactive [231]. Thus, a cell with high amounts of labile iron and disrupted ferritin function could be more sensitive to ferroptosis [232]. Many fluorescent dyes used to measure intracellular iron, such as Calcein-AM (Invitrogen C1430; Eugene, OR, USA), Phen Green SK (Invitrogen P14313; Eugene, OR, USA), and CP655, cannot distinguish between Fe^2+^, Fe^3+^, and other metal ions present in the cell (Table 2) [233]. The fluorescence intensities of these probes are quenched as they bind to and chelate intracellular iron [233,234]. FerroOrange (Dojindo Laboratory F374; Kumamoto, Kyushu, Japan) is a commercially available fluorescent dye that is specific for labile iron Fe^2+^ without chelation effects [235]. In addition to FerroOrange, Li et al. and Hirayama et al. generated fluorescent probes that specifically bind to Fe^2+^ [236,237].

### 5.5. Drawbacks and Considerations for Fluorescent Dyes

While these dyes stain their indicated organelles/targets of interest with minimal background and provide a metabolic readout, one consideration is that these reagents are exogenous and may cause toxicities or changes in metabolic phenotype upon entering the cell. For example, some of the fluorescent dyes that assess iron metabolism are chelation-based probes [233,238]. If cells are incubated in chelation-based probes for an extensive period of time, then all of the free iron in the cell will be chelated and ultimately lead to cell death. The MitoTracker probes contain a chloromethyl ester group that binds to proteins inside the mitochondria after accumulation, allowing them to stay inside the organelle. This can cause alterations in mitochondrial function in just a few hours [239]. If timelapse imaging needs to be performed over a period of hours or days, however, it is recommended to not utilize these types of probes due to potential toxicities from prolonged exposure. For this context, we recommend using stable-expressing fluorescent probes that have a lower impact on cell viability and metabolic phenotype over time, such as genetically encoded fluorescent biosensors (Figure 1).

**Table 2 ijms-26-08466-t002:** Summary of fluorescent probes to assess glucose metabolism, mitochondrial function, lipid metabolism, and iron metabolism.

Metabolism	Fluorescent Dye	Mechanism of Action	Compatible with Fixation	Reference
Glucose	2NBDG	Enters the cell via GLUT transporters and becomes phosphorylated	No	[162,163]
	6AzGal + BDP-DBCO	6AzGal enters the cell via GLUT transporters → click-chemistry to tag fluorophore to 6AzGal	No	[168]
Mitochondrial	Tetramethyl rhodamine methyl ester (TMRM)	Accumulates into mitochondria via negative membrane potential; sensitive to changes in membrane potential after staining	No	[180,187]
	Tetramethyl rhodamine ethyl ester (TMRE)	Accumulates into mitochondria via negative membrane potential; sensitive to changes in membrane potential after staining	No	[180,240]
	JC-1	Forms aggregates (red emission) in mitochondria with negative membrane potential; reverts back to monomeric state (green emission) upon mitochondrial depolarization	No	[190]
	MitoTracker Orange/Red/Deep Red	Accumulates into mitochondria via negative membrane potential and forms covalent bond with proteins within the mitochondrial matrix with a thiol-reactive functional group	Yes	[239]
	MitoTracker Green	Passive diffusion into mitochondria and forms covalent bond with proteins within mitochondrial matrix with a thiol-reactive functional group	No	[239]
	10-N-nonyl acridine orange (NAO)	Binds to cardiolipin in the mitochondria	No	[183]
Lipid	BODIPY 558/568 C12	Diffuses through the plasma membrane and is trafficked or incorporated into various organelles. Mainly used to trace where fatty acids localize to in the cell	Yes	[204]
	Nile Red	Solvatochromic dye that only becomes fluorescent in a hydrophobic environment, differential emission when bound to polar lipids (i.e., phospholipids) and neutral lipids (lipid droplets)	Yes	[208,211,212]
	Lipi-Blue/Red/Green	Only becomes fluorescent in hydrophobic environments through pyrene and perylene ring structures, but it is specific for neutral lipids in lipid droplets and no other lipid species	Yes	[210]
	BODIPY 493/503	Diffuses through the plasma membrane and incorporates into lipid droplets by interacting with neutral lipids, which are most prevalent in lipid droplets	Yes	[209,210]
	Probe 10	Coumarin attached to a fatty acid; when fully cleaved by all steps of the FAO pathway in the mitochondria, coumarin fluorescence is activated. Can be used to assess FAO flux by taking timelapse images and observing increasing fluorescence intensity over time	No	[215]
	Diphenylhexatriene (DPH)	Fluorescent lipid moiety that is quenched upon lipid peroxidation; increase in fluorescence decay indicates an increase in the rate of lipid peroxidation	No	[241]
	Liperfluo	Becomes fluorescent when oxidized by lipid hydroperoxides and peroxyl radicals	No	[219]
	BODIPY-C11	Ratiometric dye to detect undamaged lipids and oxidized lipids. Fluorescence emission changes from red to green as lipid peroxidation occurs	Yes	[219,220,221,242]
Iron	Calcein-AM	Fluorescence quenching when bound to Fe^2+^, Fe^3+^, Ni^2+^, Cu^2+^, Co^2+^	No	[233,234]
	Phen Green SK	Fluorescence quenching when bound to Fe^2+^, Fe^3+^, Ca^2+^, Zn^2+^	No	[233,234]
	CP655	Fluorescence quenching when bound to Fe^2+^, Fe^3+^, Cu^2+^	No	[233,238]
	FerroOrange	Specifically binds to Fe^2+^ irreversibly. Does not react with Fe^3+^ or chelated iron	No	[235]
	BDP-Cy-Tpy	Ratiometric probe; when bound to Fe^2+^, fluorescence intensity of Cy-Tpy fluorophore is quenched while BDP fluorescence remains the same. Increase in ratio of BDP to Cy-Tpy intensity indicates the presence of Fe^2+^	No	[236]
	RhoNox-1	Not fluorescent in oxidized state, but exhibits rhodamine-based fluorescence when reduced by Fe^2+^	No	[237]

## 6. Genetically Encoded Fluorescent Biosensors

Genetically encoded fluorescent biosensors are powerful tools to visualize the dynamics of biological processes and interactions in living cells [243]. Using stably expressed fluorescent proteins tagged to the molecules or proteins of interest, these processes can be observed in real time and provide opportunities for long-term readouts [244] (Figure 1). These sensors can be engineered to localize in certain compartments of the cell, such as the nucleus, mitochondria, endoplasmic reticulum, or the lysosome [245]. Given that these biosensors are primarily detected with one wavelength, they offer the opportunity for multiplexing with other sensors or with fluorescent dyes to obtain more information on other metabolic reactions or localization of the fluorescent probe [243,245]. This can be a powerful way to study reactions that affect one another or how a metabolite moves to an organelle for downstream signaling in response to stimulus. One advantage of using fluorescent biosensors is the ability to capture the kinetics of reactions that have high turnover rates, especially those involving metabolites that contribute to multiple pathways and have short half-lives such as acetyl Coenzyme A (acetyl-CoA) [68,244,246]. Genetically encoded biosensors are typically set up to have a sensing component that is sensitive to a metabolite, protein interaction or activity that changes its conformation to interact with the reporter component [243]. The reporter emits a fluorescence signal when excited by a specific wavelength of light and depending on the type of reporter or detection technique, the fluorescence intensity or signal decay would be proportional to metabolic activity [247]. The most common fluorescent biosensors are based on Förster resonance energy transfer (FRET) [243]. FRET occurs when two fluorescent proteins (a donor and an acceptor) are in proximity of one another, and excitation of the donor fluorophore transfers its emission energy to excite the acceptor fluorophore [248]. Thus, when an analyte binds to the sensing domain, it changes the conformation of the donor fluorophore to be in close proximity to the acceptor fluorophore [245]. When excited at the wavelength of the donor fluorophore, the emission energy is transferred to the acceptor, and the emission from the acceptor fluorophore is detected [248]. If there is no stimulus or analyte to induce the conformational change, the excitation of the donor fluorophore will only result in detection of the emission of the donor fluorophore. This fluorescence change based on conformational change and distance provides a dynamic detection range for metabolic activity, as ratiometric measurements can be made between the fluorescence intensities of the donor fluorophore and the acceptor fluorophore [245,248]. Another type of fluorescent biosensors is intensiometric, which is based on changes in fluorescence intensity of a single fluorophore upon interaction with an analyte [243,249]. In this section, we will summarize ratiometric and intensiometric biosensors that are used to study metabolic activity. In addition, there are multiple techniques to detect signals from genetically encoded fluorescent biosensors, such as fluorescence microscopy, two-photon microscopy, and fluorescence lifetime imaging microscopy (FLIM) [250,251]. FLIM imaging is dependent on an inherent attribute of the fluorophore called the fluorescence lifetime, which is the average amount of time an excited fluorophore takes to exponentially decay to a resting state [252]. In the context of sensors and reporters, the fluorescence lifetime of a fluorophore changes when bound to the analyte, allowing for the detection of that analyte [253]. The magnitude of the change in fluorescence lifetime is proportional to the amount of analyte in the system [252,253]. The different types of fluorescent biosensors are listed, along with their detection method, in Table 3.

### 6.1. Glucose Sensors

There are many published fluorescent glucose sensors in the field using either FRET or single-fluorescent-protein designs. FLIPglu-600µ was one of the first glucose FRET biosensors developed using a bacterial D-glucose-galactose-binding periplasmic protein (MglB) with Cyan Fluorescent Protein (CFP) and Yellow Fluorescent Protein (YFP) linked to it [254]. MglB changes conformation when glucose binds to it, allowing CFP to move in close proximity to its FRET pair, YFP, and YFP emission is detected when excited at the wavelength for CFP [254]. This sensor was able to measure glucose dynamics in COS-7 and HepG2 cells but lacked dynamic range in fluorescence intensity detection over the background signal [254,255,256]. Takanaga et al. built on the framework of the FLIPglu-600µ sensor and generated FLII12Pglu-700µδ6 [256]. This sensor uses Citrine-YFP as the acceptor fluorescent protein rather than the eYFP protein, which is sensitive to pH [256]. In addition, they made modifications to the placement of the fluorescent proteins relative to the MglB glucose-binding protein, allowing for greater energy transfer and FRET signal detection [256]. This allowed for FLII12Pglu-700µδ6 to achieve a dynamic range of detecting glucose from 0.05 to 9.6 mM, rather than 0.074 to 6.1 mM, as seen with FLIPglu-600µ [256]. This gave FLII12Pglu-700µδ6 the ability to detect blood glucose in vivo, which is at 5 mM [256]. Ghezzi et al. utilized the glucose FRET biosensor, FLIPglu-600µ, to measure changes in the glucose uptake rate in lung cancer cells [257]. Through this and validation with various assays, they identified that Cyclin-dependent kinase 7 (CDK7) is a driver of glucose consumption and inhibition with the CDK7 inhibitor, Milciclib, decreased glucose metabolism in non-small-cell lung cancer cells [257]. Kondo et al. used the FLII12Pglu-700µδ6 FRET-based glucose biosensor to identify intra-tumor heterogeneity in metabolic states that were driven by non-genetic mechanisms in breast cancer cell lines and in vivo models [258]. Interestingly, the observed heterogeneity was heritable through mitosis [258].

Green and Red Glifons are examples of single-fluorescent-protein-based glucose sensors that have been recently developed to visualize glucose dynamics in live cells [259,260]. The developers inserted MglB between regions of a fluorescent protein with linkers—Citrine Green Fluorescent Protein (GFP) for Green Glifons and mApple for Red Glifons [259,260,261]. When glucose is present, it binds to MglB and induces a conformational change in the protein, bringing the two regions of the fluorophore closer together and increasing fluorescence intensity [259,260]. When screening for which linkers and amino acids were optimal to obtaining a dynamic range of fluorescence signal detection, the authors found that Green Glifons yielded a 3.3-fold increase in fluorescence intensity in the presence of glucose, which led them to select it as their top candidate [259]. Another single-fluorescent-protein-based glucose sensor is based on the mTurquoise2 fluorescent protein. Zhong et al. developed the qmTQ2-glucose sensor to be imaged with the FLIM platform, which eliminates a lot of the drawbacks in conventional fluorescence microscopy of single-fluorescent-protein-based biosensors such as photobleaching and sensor concentration [253]. The qmTQ2-glucose sensor consists of the MglB protein inserted into the mTurquiose fluorescent protein with two linkers and has been shown to exhibit dose-dependent changes in fluorescence lifetime with increasing glucose concentrations [253]. This sensor, and FLIM sensors in general, are advantageous in imaging over long periods of time as the readout does not rely on fluorescence intensity and therefore not affected by photobleaching [253]. For more detail on various fluorescent biosensors that detect glucose, refer to the extensive review by Li et al. [262].

### 6.2. Glycolysis Sensors

Investigating glycolysis is not limited solely to biosensors that bind to glucose. Just downstream of glucose phosphorylation via hexokinase, fructose 1,6-bisphosphate (FBP) is generated in the third step of the glycolytic pathway by phosphofructokinase (PFK) [263,264]. The production of FBP is a rate-limiting step that commits the cell to the breakdown of glucose to pyruvate, ultimately regulating glycolytic flux [264]. HYlight is a fluorescent biosensor consisting of a circular permuted GFP (cpGFP) that is integrated into a bacterial central glycolytic gene repressor (CggR) protein that binds to FBP [264]. This ratiometric sensor emits light when excited at 400 nm and 488 nm, which corresponds to FBP binding. With increasing concentrations of bound FBP, the emission from 400 nm excitation decreases, while the emission from 488 nm excitation increases [264]. The fluorescence intensity ratio of 488 nm emission to 400 nm emission provides a readout of relative FBP concentration that is independent of biosensor abundance in each cell [264]. HYlight can report FBP concentrations ranging from 1 µM to 100 µM in vitro; however, when studied in cells, HYlight was observed to also bind to glucose 6-phosphate, fructose 6-phosphate, and dihydroxyacetone phosphate (DHAP) at high concentrations, decreasing the sensor’s measured affinity for FBP [264]. HYlight was used to study glycolytic flux in a model of liver cancer cells. The authors identified that FBP was present in low micromolar concentrations in liver cancer cells, in contrary to previously hypothesized millimolar ranges [265,266]. They also found that HYlight was specific to measuring FBP production and consumption through glycolysis and not gluconeogenesis [265]. This sensor was also used to study the effects of glycolysis on cell cycle dynamics, where the authors identified that mTOR-driven inactivation of anaphase promoting complex/cyclosome (APC/C) in quiescent cells increases glycolytic flux to promote re-entry into the cell cycle, leading to a G0 to G1 transition [267]. Given that cancer cells exhibit dysregulated metabolism and cell cycle dynamics, these findings provide a new molecular network for therapeutic targeting [267].

Pyruvate is the major end product of glycolysis and a key node that connects various pathways in central carbon metabolism [268]. Pyruvate can be fermented to lactate and excreted out of the cell, be shuttled into the mitochondria via the mitochondrial pyruvate carrier (MPC) to generate acetyl-CoA, converted to oxaloacetate through pyruvate carboxylase (PC) to replenish carbon sources in the TCA cycle, or transaminated to alanine for amino acid and protein synthesis [79,80,81,268]. Pyronic was the first fluorescent biosensor developed to detect intracellular pyruvate [269,270]. This FRET-based sensor consists of the monomeric teal fluorescent protein (mTFP) and Venus fluorescent protein flanking the pyruvate dehydrogenase complex repressor (PdhR) protein from *Escherichia coli* [269]. Glycolytic pyruvate production was measured through Pyronic and validated with glucose starvation and inhibition of pyruvate importers [269]. In addition, this sensor was able to assess the relative flux of mitochondrial pyruvate consumption when incubating glucose-deprived cells with exogenous pyruvate and monitoring the clearance of imported pyruvate over time [269]. To improve the dynamic range of the original Pyronic sensor, a single-fluorophore version was generated as PyronicSF [270]. This sensor contains the same PdhR protein but is linked to a cpGFP instead of the FRET pair, allowing for a single excitation at 488 nm to detect the signal from pyruvate and opening up channels for multiplexing with other dyes and markers [270]. PyronicSF exhibited an increase greater than a 6-fold in fluorescence intensity compared to the original Pyronic and was used to quantify pyruvate concentrations in the mitochondria [270]. Due to its smaller size relative to Pyronic, PyronicSF can be targeted to various organelles of the cell with higher efficiency and report on pyruvate dynamics within those compartments [270].

Lactate is a byproduct of glycolytic metabolism [271]. Cancer cells, through the Warburg effect, take in large amounts of glucose and break it down through glycolysis, leading to an increase in lactate excretion into the extracellular space through monocarboxylate transporter 4 (MCT4) activity [3,4,123]. In addition, cancer cells can take in lactate through monocarboxylate transporter 1 (MCT1) and use it to fuel the TCA cycle and various signaling processes [123]. Nasu et al. developed an intensiometric L-lactate sensor, eLACCO1.1, to help dissect lactate transport in mammalian cells [272]. This sensor consists of the bacterial TTHA0766 L-lactate-binding periplasmic protein linked to the cpGFP reporter [272]. The authors were able to image the release of lactate into the extracellular space from glycolytic metabolism in glioblastoma cells [272]. They targeted the eLACCO1.1 sensor to the cell surface and observed an increase in fluorescence intensity as the cells were cultured in 25 mM glucose [272]. When treated with MCT1 inhibitors, the fluorescence intensity in presence of 25 mM glucose decreased, indicating that the sensor was reporting lactate that was specifically excreted from that transporter [272]. The authors further improved their biosensor to increase the fluorescence intensity 1.38-fold with eLACCO2.1 in mammalian cells [273]. In addition, they also engineered a sensor to detect levels of intracellular L-lactate called R-iLACCO1 [273]. This sensor contains a circular permuted monomeric Apple (cpmApple) red fluorescent protein that is linked to a bacterial L-lactate binding protein, LldR [273,274]. Treating glucose-starved cells with glucose increased the fluorescence intensity of the R-iLACCO1 sensor, reflecting the activation of glycolysis and subsequent lactate production in the cell [273]. Other lactate sensors in the field include FiLa, FiLa-Red and LiLac [244,275,276].

### 6.3. Acetyl-CoA Sensors

Acetyl-CoA is an intermediate metabolite in central carbon metabolism that bridges glycolysis, the TCA cycle, and fatty acid synthesis [277,278]. In addition, it has been shown to be used for signaling and post-translational modification of histones [277,279]. Due to the various functions of acetyl-CoA, Smith et al. developed the PancACe fluorescent biosensor to track the localization and presence of acetyl-CoA in living cells and their compartments [280]. They engineered this sensor with a bacterial acetyl-CoA-binding protein, PanZ, and a cpGFP fluorescent protein [280,281]. With this framework, the fluorescence of cpGFP is activated as acetyl-CoA binds to PanZ and changes its conformation [280]. Other than PancACe, there are no acetyl-CoA-specific fluorescent biosensors for live cell imaging. Liberman et al. developed a bioluminescence resonance energy transfer (BRET)-based sensor to detect acetyl-CoA, but that is limited to cell lysates and not compatible with a live cell imaging platform [280,282]. Since cancer cells have been shown to alter lipid metabolism, histone acetylation, and mitochondrial metabolism, using sensors to track how acetyl-CoA play a role in these phenotypes in a spatiotemporal context can provide potential therapeutic targets [280,283].

### 6.4. Redox Metabolism Sensors

Redox metabolism is a critical factor contributing to therapy resistance and disease progression in cancer, as it helps balance the levels of reactive oxygen species (ROS) in the cell and modulates other metabolic pathways to sustain proliferation [284]. Two of the most studied redox metabolites are NADH and its oxidized form, NAD+. These metabolite acts as electron carriers for various redox reactions in central carbon metabolism that occur in the cell [285]. For example, NAD+ is reduced to NADH when the glycolytic metabolite, 3-phosphoglycerate (3PG), is oxidized for serine biosynthesis [285]. NAD+ is also reduced to NADH in the mitochondria during multiple reactions in the TCA cycle [285]. The NADH produced from the TCA cycle is then utilized in oxidative phosphorylation by Complex I (NADH dehydrogenase) [285]. Complex I oxidizes NADH to NAD+, transferring the electrons to ubiquinone (CoQ) and further down the ETC to generate the proton gradient for ATP production [285]. The balance of NADH and NAD+ is imperative for maintaining metabolic homeostasis in the cell; so, the ratio of these two metabolites have been extensively studied. Hung et al. developed a fluorescent biosensor to image the cytosolic NADH/NAD+ metabolites called Peredox [286]. They engineered a cpT-Sapphire fluorophore between two subunits of the bacterial NADH binding protein, Rex, and attached an mCherry fluorophore downstream of the second subunit of Rex [286,287,288]. When Peredox is bound to NAD+, the cpT-Sapphire does not emit green fluorescence; so, only the red signal from mCherry is detected [286]. When Peredox is bound to NADH, the conformational change allows for cpT-Sapphire to emit green fluorescence in addition to detecting the red signal from mCherry [286]. Thus, the NADH to NAD+ ratio can be calculated as the ratio of green to red fluorescence intensity [286]. Birts et al. applied this sensor to report that high NADH/NAD+ ratios increase the stability of p53 to sustain glycolytic activity in cancer cells [289]. Another NADH/NAD+ fluorescent biosensor is based on a circularly permuted YFP (cpYFP) reporter on Rex called SoNar [290]. When bound to NAD+, the sensor predominantly emits fluorescence when excited at 485 nm but not at 420 nm [290]. When bound to NADH, the fluorescence intensity ratio increases when excited at 420 nm and 485 nm, indicating an increase in the NADH/NAD+ ratio [290]. Using this sensor, the authors were able to screen over 5000 compounds from 23 libraries to identify which ones impair the NADH/NAD+ redox state in cancer cells [290]. They identified 78 compounds that increased the NADH/NAD+ ratio and 12 compounds that decreased the NADH/NAD+ ratio [290]. From those cutoffs, they then narrowed down to KP-372-1, a known Akt inhibitor, which exhibited the largest fold decrease in the NADH/NAD+ ratio and induced the most cytotoxicity in vitro in the nanomolar range [290]. KP-372-1 was also efficacious in reducing tumor growth in vivo and exhibited good pharmacokinetic parameters, such as a long half-life and high bioavailability [290]. The SoNar sensor is also responsive to subtle changes in the NADH/NAD+ ratio and has a broad dynamic range in signal intensity, making it useful for drug screening for compounds to target cell metabolism [290].

Another well-studied redox metabolism pair is the ratio of reduced glutathione (GSH) to oxidized glutathione (GSSG). Glutathione is a tripeptide that is composed of glutamate, cysteine, and glycine [291]. It plays major roles as an antioxidant to reduce levels of reactive oxygen species in cells and can act as a chelating agent for metals to prevent metal toxicity [291,292]. GSH is also important for replenishing the activity of glutathione peroxidase in lipid membranes (GPX4) to prevent ferroptosis [293]. Thus, measuring the GSH/GSSG ratio in cancer cells can reveal their state of oxidative stress and consequently predict their potential susceptibilities to ROS-inducing and antioxidant-depleting therapies such as GPX4 inhibitors and system XC inhibitors [293]. One of the first GSH/GSGG fluorescent biosensors developed was Grx1-roGFP2, which consisted of a redox-sensitive GFP reporter (roGFP) fused to glutaredoxin-1 (Grx1) [294]. Grx1 alters the redox state of the roGFP reporter in the presence of GSSG, ultimately changing the fluorescence intensity of roGFP [294]. This sensor provides a ratiometric readout of GSH/GSSG based on fluorescence intensities when excited at 390 nm and 480 nm [294]. When bound to GSSG, the roGFP reporter emits a fluorescence signal, and the activity of Grx1 alters the oxidation state of roGFP to enhance the fluorescence signal even further and at a faster rate than roGFP alone, allowing for rapid detection of changes in GSH/GSSG [294]. This sensor was able to measure the intracellular glutathione ratios in HeLa cells cultured in different environmental conditions, as well as changes in those ratios in TRAIL-induced apoptosis [294]. The sensor was further improved into Grx1-roGFP2.iL to expand the dynamic range of the roGFP reporters in response to GSH and GSSG [295]. Abbas et al. designed a fluorescent biosensor to detect compartment-specific GSH/GSSG ratios and applied it to studying the antioxidant response in acute myeloid leukemia (AML) cells under chemotherapy [296]. They based their designs off of the previously discussed reporters, Grx1-roGFP2 and Grx1-roGFP2.iL, and tagged localization signals onto them [294,295,296]. They generated a cyto-Grx1-roGFP2 to localize in the cytoplasm, an MLS-Grx1-roGFP2 to localize to the mitochondria, and an NLS-Grx1-roGFP2 to localize to the nucleus [296]. With the compartment-specific sensors, the authors observed that the GSH/GSSG ratio dropped the most in the mitochondria and nucleus in response to glutaminase inhibition with CB-839 [296]. The application of this sensor allows the user to identify compounds that alter redox status in specific organelles in the cell, which can give rise to context-specific therapeutic vulnerabilities against cancer.

### 6.5. Adenosine Triphosphate (ATP) Sensors

ATP is an essential resource for cells to maintain energetic homeostasis [297]. It is involved in numerous biochemical reactions to store and release energy, cell signaling processes, cytoskeletal movement, and macromolecule biosynthesis—all of which cancer cells utilize for survival, proliferation, and metastasis [297,298,299]. ATeam is one of the first biosensors generated using a microbial F_0_F_1_-ATP synthase epsilon subunit that binds ATP [300]. ATeam is a FRET-based sensor equipped with a monomeric super enhanced cyan fluorescent protein (mseCFP) as the donor fluorophore and a circular permuted monomeric Venus (cpmVenus) fluorescent protein as the acceptor fluorophore [300]. Lobas et al. then adopted the microbial F_0_F_1_-ATP synthase epsilon subunit and inserted a cpGFP into its α-helices to create iATPSnFR^1.0^ and iATPSnFR^1.1^ as a single-wavelength biosensors to measure ATP in the cytosol and the extracellular space [301]. They can specifically detect ATP with a dynamic range of 30 µM to 3 mM and are not sensitive to AMP, ADP, or adenosine [301]. In addition, they can be targeted to various cellular compartments and the cell membrane to obtain organelle-specific and cell surface-specific ATP dynamics, respectively [301]. In various breast cancer cell lines, the iATPSnFR sensor revealed a novel spatiotemporal modulation of glycolysis-derived ATP correlating with waves of energy-costly activities such as migration and macropinocytosis at the plasma membrane [302].

Another readout of ATP metabolism is the ratio of ATP to ADP, which dictates the cell’s energetic status and regulates the relative activity of glycolysis and mitochondrial metabolism [303,304]. A widely used sensor for detecting the ATP/ADP ratio is PercevalHR, which consists of a bacterial ATP-binding protein, GlnK1, with several amino acid residue mutations in its ATP-binding pocket, integrated with a cpmVenus fluorescent protein [304,305]. This ratiometric-based sensor is based on the fluorescence signal emitted via excitation at two different wavelengths that correspond to ATP or ADP binding [304]. ADP binding increases fluorescence intensity from 405 nm excitation, while ATP binding increases fluorescence intensity from 488 nm excitation [304]. The ratio of those two signals allows for the relative quantification of ATP/ADP that is independent of the amount of biosensor in the cell [304]. In studies investigating cell motility in TNBC, PercevalHR was used to capture the heterogeneity of ATP/ADP ratios in MDA-MB-231 cells exposed to different extracellular matrix environments [306,307]. The authors identified that ATP production and consumption are dependent on the demand of cytoskeletal components to move, and that highly motile cancer cells increase their ATP/ADP ratios in response to increasing collagen density, while cells with low motility did not [306,307]. These findings further demonstrate that cellular energetics play an essential role in metastasis [306,307].

### 6.6. Drawbacks and Considerations for Genetically Encoded Fluorescent Biosensors

One drawback of this approach is the need to generate stable cell lines. Since this approach uses fluorescent proteins fused to one’s targets of interest, the metabolic phenotype or growth rate should be assessed in the cells post-transduction. If the plasmid construct contains a highly active promoter, such as the human cytomegalovirus (CMV) promoter, one’s fluorescently tagged protein of interest should be overexpressed and could impair or augment the metabolic phenotype of one’s cells. In addition, one should assess whether the expression of the fluorescent proteins do not impair the function of the target of interest, as some fluorescent proteins are large and may sterically hinder domains for catalysis or protein-protein interactions. It may also be important to generate single-cell clones to ensure uniform and homogenous activity of the fluorescent probes at the population level (especially for FRET-based experiments). In addition, it is imperative to assess the pH sensitivity of the fluorescent biosensor, as acidic conditions have been shown to affect fluorescence of GFP reporters [308]. For example, the iATPSnFR ATP sensors exhibit decreased efficiency in an environment with a pH lower than 6.2 [301].

Recent developments in optogenetic tools have also increased the versatility and impact of genetically encoded fluorescent biosensors, allowing researchers to actively manipulate a biological process and visualize the effects to investigate causal links at a spatiotemporal level [309,310,311]. These tools consist of engineered photoreceptors derived from bacteria and algae tagged to proteins of interest in cells, leading to the production of a tunable system that can alter protein activity with light exposure [309,310,311]. Zhan et al. utilized this approach to induce translocation of glycolytic enzymes from the cytoplasm to the cell surface in real-time and observed an increase in cancer cell migration [302]. Tkatch et al. introduced a channel rhodopsin into the inner mitochondrial membrane to induce depolarization of the mitochondrial membrane potential and observed decreased mitochondrial calcium signaling [312]. This study provided a method to specifically induce mitochondrial stress without the addition of mitochondrial poisons, which do not act reversibly or at the single-cell level [312]. Utilizing optogenetic tools with genetically encoded fluorescent biosensors to activate or perturb specific metabolic enzymes and study the downstream phenotypic effects in cancer cells may shine light on novel molecular interactions that can be therapeutically targeted.

If one is looking to generate new fluorescent biosensors, it is important to perform screens with mutagenesis in the components of the starting biosensor structure to generate the best-acting fluorescent biosensor based on the affinity of the sensor for the analyte and the kinetics and range of the fluorescent signal [245,256]. Koveal et al. provides a workflow for high-throughput screening to develop these tools to help speed up the process [244]. It may also be beneficial to consider developing high-throughput image analysis pipelines to obtain spatiotemporal resolution of the metabolic readouts from the biosensors at the single-cell level. Wollamn et al. developed a MATLAB (R2025a)-based image analysis pipeline, FRETzel (Version 1.0), to segment cells and measure FRET-based glucose intensity ratios and has tested it in various cell types such as adipocytes, yeast, and fibroblasts [313]. Other software that can be used to analyze data from FRET experiments at a single-cell level includes AccPbFRET (Version 3.15), FLIM-FRET Analyzer (Version 0.3.0), and CellProfiler (Version 4.2.8) [313,314,315,316].

**Table 3 ijms-26-08466-t003:** Different genetically encoded fluorescent biosensors used to assess glucose, FBP, pyruvate, lactate, acetyl-CoA, redox, and ATP metabolism.

Type of Sensor	Sensor Name	Fluorescent Protein(s)	Ratiometric, Intensiometric, or Lifetime; Method of Detection	Reference
Glucose	FLIPglu-600µ	eCFP & eYFP	Ratiometric; FRET	[254]
	FLII12Pglu-700µδ6	eCFP & Citrine-YFP	Ratiometric; FRET	[256]
	Green Glifon	Citrine-GFP	Intensiometric; fluorescence microscopy	[259]
	Red Glifon	mApple	Intensiometric; fluorescence microscopy	[260]
	qmTQ2-glucose	mTurquoise2	Intensiometric with fluorescence microscopy; lifetime with FLIM	[253]
FBP	HYlight	cpGFP	Intensiometric with fluorescence microscopy	[264]
Pyruvate	Pyronic	mTFP & Venus	Ratiometric; FRET	[269]
	PyronicSF	cpGFP	Intensiometric with fluorescence microscopy	[270]
Lactate	eLACCO1.1	cpGFP	Intensiometric with fluorescence microscopy; ratiometric with two-photon microscopy	[272]
	eLACCO2.1	cpGFP	Intensiometric with fluorescence microscopy; ratiometric with two-photon microscopy	[273]
	R-iLACCO1	cpmApple	Intensiometric with fluorescence microscopy; ratiometric with two-photon microscopy	[273]
	FiLa	cpYFP	Ratiometric; fluorescence microscopy	[276]
	FiLa-Red	cpmApple	Ratiometric; fluorescence microscopy	[275]
	LiLac	mTurquoise2	Intensiometric with fluorescence microscopy; lifetime with FLIM	[244]
Acetyl-CoA	PancACe	cpGFP	Ratiometric; fluorescence microscopy	[280]
NADH/NAD+	Peredox	cpT-Sapphire & mCherry	Ratiometric: fluorescence microscopy	[286]
	SoNar	cpYFP	Ratiometric; fluorescence microscopy	[290]
Reduced/Oxidized Glutathione (GSH/GSSG)	Grx1-roGFP2	roGFP	Ratiometric; fluorescence microscopy	[294]
	Grx1-roGFP2.iL	roGFP	Ratiometric; fluorescence microscopy	[295]
ATP	ATeam	mseCFP & cpmVenus	Ratiometric; FRET	[300]
	iATPSnFR	cpGFP	Intensiometric with fluorescence microscopy	[301]
	PercevalHR	cpmVenus	Ratiometric; fluorescence microscopy	[304]

## 7. Concluding Remarks

Cancer cells alter their metabolism to sustain growth, produce precursors for macromolecule biosynthesis, and respond to redox and environmental stress, leading to resistance against standard-of-care therapies. The complex heterogeneity across different tumor types and within each tumor makes it difficult to treat patients with current metabolic inhibitor strategies. Although there have been many advances in the understanding of glutamine metabolism in various cancers, recent clinical trials using glutamine antagonists have resulted in mixed conclusions [317]. Thus, there is much work to be carried out in investigating cancer metabolism to identify novel targets. We have outlined the five most common and widely used methods in the field. There is no assay that is perfect for every situation—one must choose which is best to answer the question at hand. In a perfect world, all these assays and methods should be performed to obtain the metabolic profiles of the groups of interest, as they all provide distinct information about the system’s metabolism. However, with limitations such as access to instruments and reagents, funding, and time, this is often practically unfeasible. Figure 1 summarizes a decision flow chart as a guide to the best method to utilize based on the stage of the project and the desired experimental parameters. In addition, the methods in the flow chart contain icons indicating their applications to in vitro, in vivo, and/or ex vivo studies. We anticipate that this chart will expand as new approaches are developed to probe different aspects of metabolism as we push the boundaries of molecular biology, biomedical engineering, imaging sciences, and analytical chemistry.

## Figures and Tables

**Figure 1 ijms-26-08466-f001:**
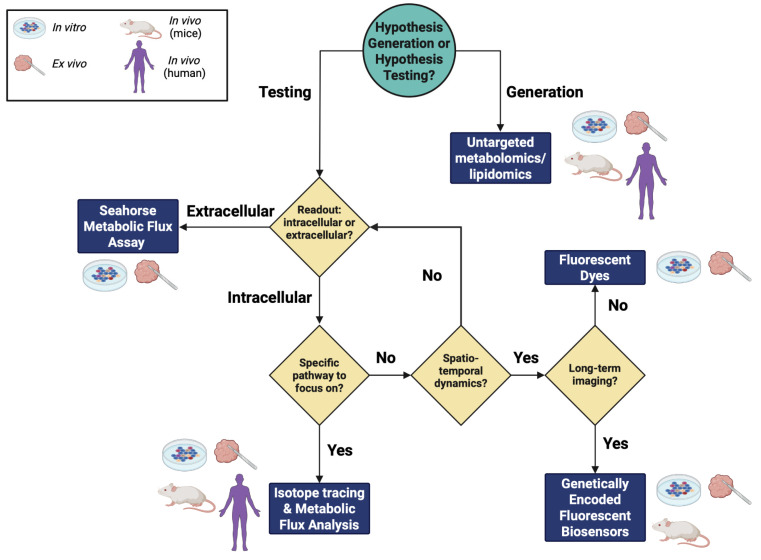
Decision flow chart for which method to use to study cancer metabolism based on experimental parameters and stage of the project. Starting at the top of the chart, one must identify if their project is hypothesis-generating or hypothesis-testing. Untargeted metabolomics/lipidomics allows for the generation of many hypotheses by providing a top–down view of the metabolome. Once a hypothesis is generated, it can be tested based on which readouts the phenotype can be detected with, such as extracellular or intracellular measurements. Extracellular measurements can be made with the Seahorse metabolic flux assay, while intracellular measurements can be performed with isotope tracing and metabolic flux analysis to test specific pathways, or with fluorescent dyes and genetically encoded fluorescent biosensors if spatiotemporal dynamics are desired. The icons next to each method indicate if they are applicable in vitro (cell culture dish), ex vivo (tissue), in vivo (mice), or in vivo (human). Created in BioRender [11].

**Figure 2 ijms-26-08466-f002:**
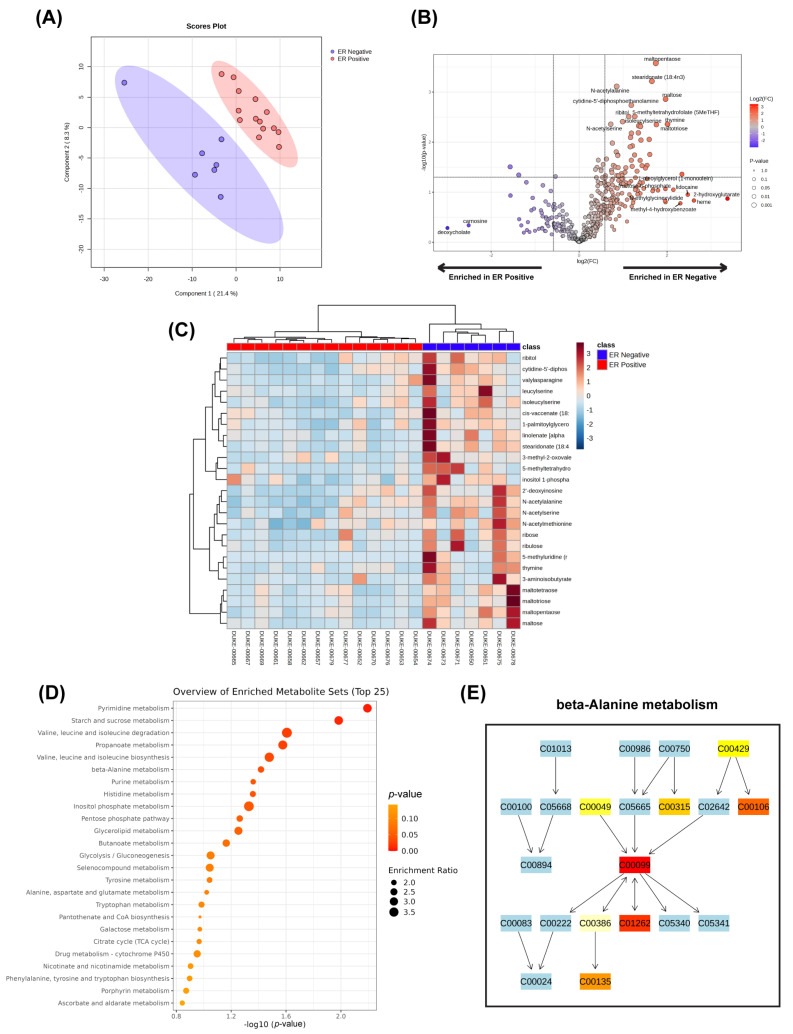
Untargeted metabolomics analysis through MetaboAnalyst 6.0. The metabolomics dataset used for these new analyses was generated by Tang et al. [18], available under a Creative Commons Attribution 4.0 International License (https://creativecommons.org/licenses/by/4.0/ (accessed on 15 July 2025)). Raw data was re-grouped by estrogen receptor expression for these analyses [18]. (**A**) Partial least squares discriminant analysis (PLS-DA) plot for dimensionality reduction between estrogen receptor-positive (ER+) and estrogen receptor-negative (ER−) breast cancer groups based on abundances of metabolites in each group. (**B**) Volcano plot indicating differentially abundant metabolites in ER+ and ER− breast cancer groups. (**C**) Heatmap of the top 50 metabolites that were altered between ER+ and ER− breast cancer groups. (**D**) Metabolite set enrichment analysis (MSEA) results based on relative abundances of metabolites in known pathways between ER+ and ER− breast cancer groups. (**E**) Pathway analysis listing individual metabolites in the beta-Alanine metabolic pathway that were altered in abundance between ER+ and ER− breast cancer groups.

**Figure 3 ijms-26-08466-f003:**
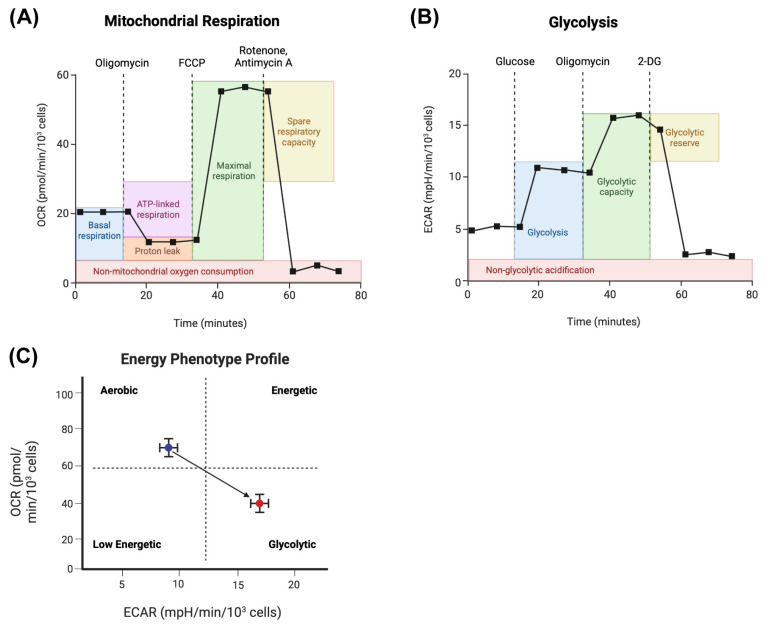
Common plots for Seahorse real-time metabolic flux analysis. (**A**) Plot of oxygen consumption rate (OCR) over time as different mitochondrial poisons are introduced to cells of tissues throughout the experiment. (**B**) Plot of extracellular acidification rate (ECAR) over time as different glycolytic agents are introduced to cells or tissues throughout the experiment. (**C**) An energy phenotype profile defined by plotting the OCR by the ECAR of a population. Aerobic phenotype consists of high OCR and low ECAR, a low-energetic phenotype consists of low OCR and low ECAR, a glycolytic phenotype consists of low OCR and high ECAR, and an energetic phenotype consists of high OCR and high ECAR. Panel A was created in BioRender [34], adapted from the “Metabolic Assays—Using Seahorse Analyzers” template. Panel B was created in BioRender [35], adapted from the “Metabolic Assays—Using Seahorse Analyzers” template. Panel C was created in BioRender [36].

**Figure 4 ijms-26-08466-f004:**
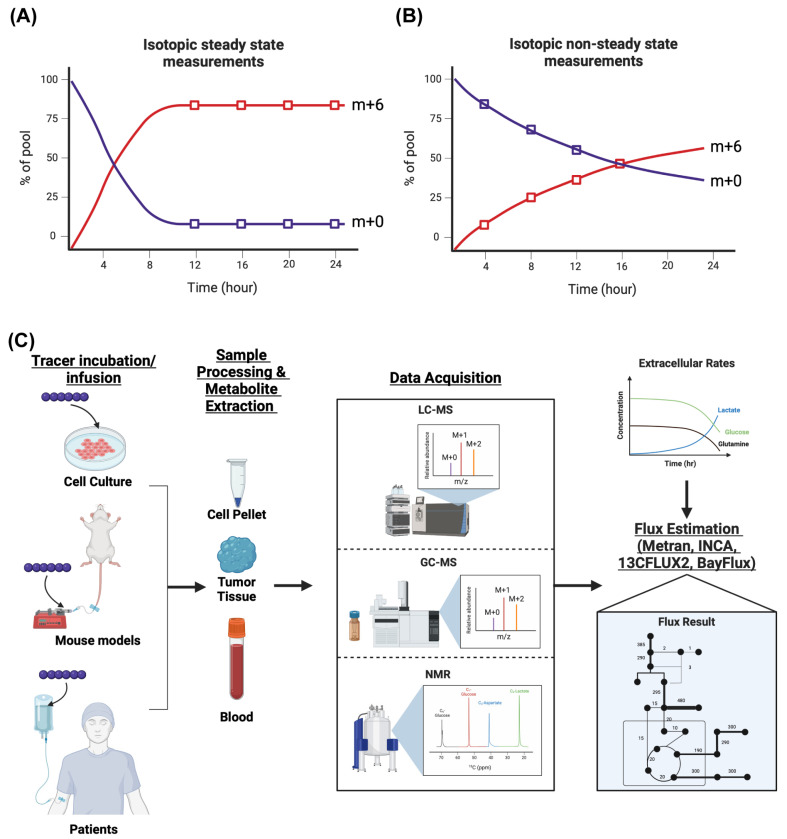
Overview of isotope-tracing methodology and workflow. (**A**) Percent labeling of [^13^C] of an arbitrary metabolite reaches steady state by 12 h. Sampling from 12–24 h can be carried out to satisfy the steady-state assumption of metabolic flux analysis. (**B**) Percent labeling of [^13^C] of an arbitrary metabolite does not reach steady state within the sampling period. Sampling can be performed at multiple timepoints to assess labeling kinetics and analyzed with isotopic non-stationary metabolic flux analysis (INST-MFA). (**C**) Isotope tracer is incubated in cells, or infused in mice or humans, followed by sample processing and metabolite extraction from cell pellets, tumor tissue, and blood. Incorporation of tracer into downstream metabolites is measured through liquid chromatography–mass spectrometry (LC-MS), gas chromatography–mass spectrometry (GC-MS), or nuclear magnetic resonance (NMR) and displayed as mass isotopomer distributions (MIDs). Data can be fitted and integrated with extracellular uptake and excretion rates for high-resolution ^13^C-metabolic flux analysis (^13^C-MFA). Panels A and B were created in BioRender [94]. Panel C was created in BioRender [95].

**Table 1 ijms-26-08466-t001:** List of natural abundance correction software and their limits.

Natural Abundance Correction Software	Programming Language	Limit for Number of Tracers to Use in Experiments	Reference
IsoCor (Version 2.2.3)	Python (Version 3.13.7)	1	[105]
PyNAC (Version 1.0)	Python (Version 3.13.7)	2	[106]
PolyMID-Correct (Version 1.0)	Python (Version 3.13.7)	1	[104]
IsoCorrectoR (Version 3.21)	R (Version 4.5.1)	2 (any combination of ^2^H, ^13^C, ^15^N, ^18^O, ^34^S)	[103]
AccuCor2 (Version 0.3.1)	R (Version 4.5.1)	2 (^2^H-^13^C or ^13^C-^15^N)	[107]
MIDcor (Version 1.0)	R (Version 4.5.1)	1	[108]
FluxFix (Version 0.1.0)	Web-based (Version 0.1.0)	1	[109]
ElemCor (Version 1.0)	MATLAB (R2025a)	1 (^2^H, ^13^C, or ^15^N)	[110]

## Data Availability

Not applicable.

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
