# Peer review of "Methods and Guidelines for Metabolism Studies: Applications to Cancer Research"

_ijms, 2025, doi:10.3390/ijms26178466_

Round 1
Reviewer 1 Report
Comments and Suggestions for Authors
Overall, this is not a review article that can be published.
- Nothing stands out from this review article compared to existing reviews.
- Three out of four figures come from one author's own article. This is extremely unacceptable for a review article, which would definitely cause biases.
- The resolution of all images is low.
- The paper is not well-organized.
Reviewer 2 Report
Comments and Suggestions for Authors
In this manuscript, the authors have comprehensively reviewed key methods useful to dissect cancer cell metabolism, including a thoughtful discussion of their limitations. I would like to congratulate the authors for a very well-written and informative manuscript, which will be valuable for both newcomers and experienced researchers in the field.
Following are my suggestions:
1) Consider including a brief discussion of some genetically encoded ATP biosensors such as iATPSnFR biosensor, PercevalHR etc. and their potential limitations.
2) The authors may add information on pyruvate biosensors such as PyronicSF and sensors for monitoring glycolysis like HYlight, etc.
3) Emerging optogenetic tools that enable spatiotemporal control of local glycolysis or mitochondrial function at a single-cell resolution can be added at the end; some examples: https://doi.org/10.1038/s41467-025-60596-6; https://doi.org/10.1073/pnas.1703623114
4) there are some minor grammatical/typographical errors throughout the manuscript. A few examples include: i) line 691: 2nbgd; ii) abbreviation of BRET; iii) line 299- MDA-MB-468; iv) line 13 (abstract) and v) line 433
Round 2
Reviewer 1 Report
Comments and Suggestions for Authors
The authors have significantly improved the article quality, which makes it a good publication.